# Engineered osteoclasts as living treatment materials for heterotopic ossification therapy

Wenjing Jin [1,2], Xianfeng Lin[3,4], Haihua Pan [5], Chenchen Zhao[3,4], Pengcheng Qiu[3,4], Ruibo Zhao[6], Zihe Hu[2], Yanyan Zhou[2], Haiyan Wu[2], Xiao Chen [7,8], Hongwei Ouyang [7,8], Zhijian Xie [2✉] & Ruikang Tang [1,5✉]

Osteoclasts (OCs), the only cells capable of remodeling bone, can demineralize calcium minerals biologically. Naive OCs have limitations for the removal of ectopic calcification, such as in heterotopic ossification (HO), due to their restricted activity, migration and poor adhesion to sites of ectopic calcification. HO is the formation of pathological mature bone within extraskeletal soft tissues, and there are currently no reliable methods for removing these unexpected calcified plaques. In the present study, we develop a chemical approach to modify OCs with tetracycline (TC) to produce engineered OCs (TC-OCs) with an enhanced capacity for targeting and adhering to ectopic calcified tissue due to a broad affinity for calcium minerals. Unlike naive OCs, TC-OCs are able to effectively remove HO both in vitro and in vivo. This achievement indicates that HO can be reversed using modified OCs and holds promise for engineering cells as "living treatment agents" for cell therapy.

[1] Department of Chemistry, Zhejiang University, Hangzhou, Zhejiang 310027, China. [2] Stomatology Hospital, School of Stomatology, Zhejiang University School of Medicine, Clinical Research Center for Oral Diseases of Zhejiang Province, Key Laboratory of Oral Biomedical Research of Zhejiang Province, Cancer Center of Zhejiang University, Hangzhou, Zhejiang 310006, China. [3] Department of Orthopaedic Surgery, Sir Run Run Shaw Hospital, Medical College of Zhejiang University, Hangzhou, Zhejiang 310016, China. [4] Key Laboratory of Musculoskeletal System Degeneration and Regeneration Translational Research of Zhejiang Province, Hangzhou, Zhejiang 310016, China. [5] Qiushi Academy for Advanced Studies, Zhejiang University, Hangzhou, Zhejiang 310027, China. [6] School of Materials Science and Engineering, Zhejiang Sci-Tech University, Hangzhou, Zhejiang 310018, China. [7] Key Laboratory of Tissue Engineering and Regenerative Medicine of Zhejiang Province, Zhejiang University School of Medicine, Hangzhou, China. [8] Dr. Li Dak Sum & Yip Yio Chin Center for Stem Cells and Regenerative Medicine, and Department of Orthopedic Surgery of the Second Affiliated Hospital, Zhejiang University School of Medicine, Hangzhou, China. ✉email: xzj66@zju.edu.cn; rtang@zju.edu.cn

In nature, calcium minerals can be readily dissolved by acids, and osteoclasts (OCs) are essential for bone resorption due to their ability to acidify extracellular microenvironments with electrogenic proton pumps[1]. OCs are multinucleated cells induced by the differentiation of monocyte/macrophage cells at or near bone. As natural bone-resorbing cells, OCs can remove large amounts of calcium, phosphate, and collagen fragments released by the dissolution of bone minerals and organic bone matrix during bone resorption, which would be toxic to cells[2,3]. Importantly, OCs are the only cells with the physiological capacity to resorb bone matrix and maintain bone content[4]. However, OCs are not directly used for ectopic calcification treatment due to their poor activity at the site of ectopically calcified tissue, such as in heterotopic ossification (HO)[5,6]. HO refers to an abnormal deposition of calcium salts formed in soft or hard tissues under pathological conditions, such as genetic mutation, trauma, or disease[7]. This condition is primarily caused by invasive surgery, tendon injury, and burns, and is typically accompanied by complications, such as limited motion and chronic pain[8]. Currently, clinical options for HO are limited, and HO is listed as a postoperative risk factor in orthopedic procedures[9,10]. Although intensive studies have investigated the underlying mechanisms of HO, such as selective retinoic acid receptor-γ[11], and BMP signaling[12–15], to prevent its occurrence, no effective methods have been developed to remove existing HO[10]. To the best of our knowledge, artificially controlled or suppressed HO has not been achieved in humans, especially using OCs.

In 2008, our laboratory proposed that yeast could be coated with calcium-phosphate shells via a layer-by-layer (LBL) strategy, which is an effective method for cell-surface modification, to enhance survivability[16]. Subsequently, cell-surface engineering has attracted a great deal of attention as a new field of research[17–20]. Increasing efforts have been made to exploit material-based modification of cells to confer other characteristics for extended use in a variety of applications, including cell protection[21], storage[22], thermal stability[23], biological stealth[17], photosynthesis and biocatalysis, etc.[24–30]. Although there are many successful strategies for the modification of living organisms, the generation of material shells with functional structures on OCs remains a challenge. Tetracycline (TC) is an organic compound commonly used in calcium-mineral targeting and has a high affinity for hydroxyapatite $Ca_{10}(PO_4)_6(OH)_2$, the major constituent of bone and other calcified tissues in humans[31,32]. TC provides three advantages: (i) the ability to covalently bond carboxyl-rich membranes of OCs[33–35], (ii) excellent biocompatibility[36], and (iii) fluorescent properties to provide easy detection[37].

In this work, we propose a rational modification of OCs to confer the ability to adhere and target ectopic calcification sites with a native biological decalcification function using TC (Fig. 1a). In vitro and in vivo experiments demonstrate that such engineered OCs are more effective than the native ones in resorbing ectopic calcifications and can be developed as therapeutic cells to remove different types of HO. This achievement shows promise for the use of chemical cells as "living treatment materials" or even "living agents" for biomedical applications.

## Results

**Osteoclast differentiation and modification.** In our study, OCs were induced by bone marrow-derived monocyte–macrophage precursor cells (BMMs) stimulated by receptor activator of nuclear factor-κB ligand (RANKL, 50 μg/l) and macrophage colony-stimulating factor (M-CSF, 25 μg/l);[38] OCs obtained in vitro were characterized by tartrate-resistant acid phosphatase (TRAP) staining (Fig. 1b), and their decalcification capacity

(Supplementary Fig. 1) and bone-targeting ability were confirmed. As expected, their adhesion ability within ectopic calcified tissue and soft tissues was poor (Supplementary Fig. 2), which was also confirmed by previous study[5,6]. Our results showed that OCs cultured in the presence of TC were characterized by TC cocoons, these cells were named TC-engineered osteoclasts (TC-OCs) in this study. We determined an optimized strategy for treating $2 \times 10^5$ OCs/ml with 160 μg/ml of TC, yielding a coating efficiency of approximately $90.7 \pm 2.0\%$ and a cell viability of $89.0 \pm 4.9\%$ (Fig. 1c–e and Supplementary Fig. 3). As shown in the Fourier transform-infrared (FTIR) spectrum, the intensities of the characteristic peaks of C–N bonds (approximately $1260\,cm^{-1}$), carbonyl-stretching vibrations (approximately $1680\,cm^{-1}$) and N–H-stretching vibrations (approximately $3000\,cm^{-1}$) increased after modification, indicating that the conjugation of amino groups on TC and carboxyl groups on the cell membrane formed more amide bonds (Fig. 1f). Furthermore, the surface-modification process was also detected by Raman spectroscopy, and the performance of the TC peak ($1639\,cm^{-1}$) indicated the successful modification of the cell surface (Supplementary Fig. 4a).

Under a confocal laser-scanning microscope (CLSM), a coated TC (green) layer was tightly associated with each cell membrane (red, labeled by PKH26); naive OCs had no such green layer (Fig. 1g). Multiple nuclei (blue, labeled by Hoechst 33258) were observed in encapsulated cells, confirming the successful chemical modification of TC on the OCs, and the fluorescence intensity of OCs and TC-OCs as determined by fluorescence spectrophotometry was consistent with these results (Supplementary Fig. 4b). The results showed that the TC cocoons were stable on the cells over four days (Fig. 1g and Supplementary Fig. 4c). A quantitative estimation indicated that each cell was modified with approximately $1.35 \times 10^{11}$ TC molecules (Supplementary Fig. 4d) (see the supplementary file for the detailed calculation process).

The ruffled membranes on the OC cell surface form a specialized region that acidifies calcium minerals in a resorption space[39,40]. Accordingly, we assessed whether this primary characteristic of OCs was affected by TC modification. As shown by scanning electron microscopy (SEM), no differences were observed between the OCs and TC-OCs in the acid-etched areas on bone tissues (Supplementary Fig. 5a, b). Furthermore, examination by inductively coupled plasma optical emission spectrometry (ICP-OES) showed that the amounts of $Ca^{2+}$ released from the acid-etched areas into the culture medium in the two groups were the same (Supplementary Fig. 5c). The OCs/TC-OC resorption site on the cortical bone surface was quantitatively and qualitatively confirmed by transmission electron microscopy (TEM) and atomic force microscopy (AFM) (Fig. 1h, i and Supplementary Fig. 5d and Supplementary Fig. 6). The depth of the pit was approximately 20 μm, which confirmed that cells had excellent bone-resorption capacity. These results indicated that the TC-OCs and OCs shared the same calcium-mineral resorption capacity, confirming that the fundamental function of OCs remained after TC modification.

**Functional verification of TC-OCs.** The Transwell migration assay is a common technique for studying cell migration behavior in vitro[41]. OCs and TC-OCs ($3 \times 10^4$/well) were seeded onto permeable filter inserts, which were placed in the wells of culture plates and in direct contact with medium containing ectopic calcified tissue (Fig. 2a, b). Within a period of 12 h, the migration ability of TC-OCs onto the tissues in the lower chamber increased by more than threefold compared with that of naive OCs (numbers: OCs: $80 \pm 10$; TC-OCs: $355 \pm 21$) (Fig. 2c–g). The results showed that the cell-migration capacity improved as

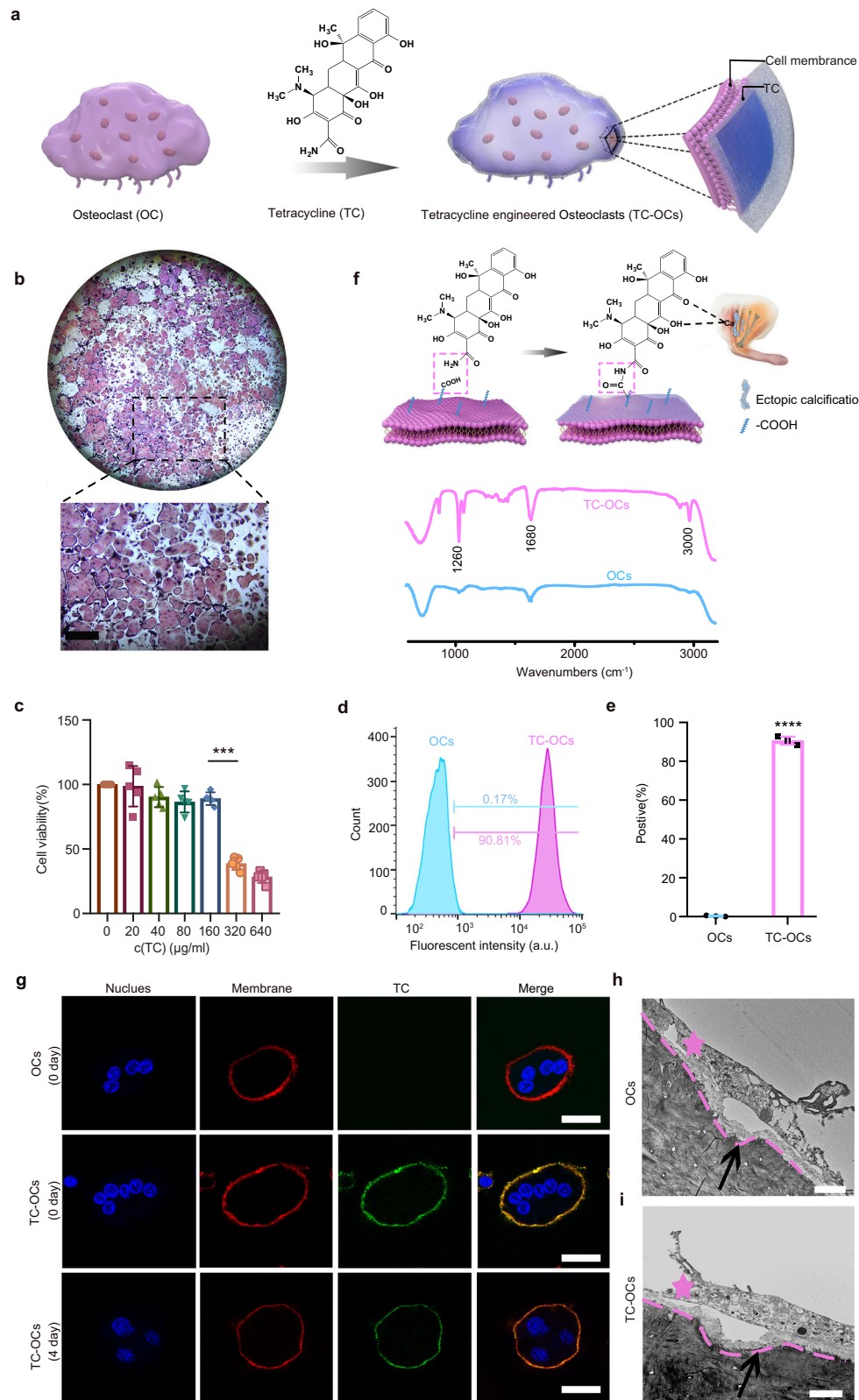

the TC concentration increased (Supplementary Fig. 7). This improved targeting effect to the ectopic calcified sites was due to the surrounding TC cocoons.

In addition, the enhanced attractive forces between the TC-OCs and calcified tissue were quantitatively confirmed by AFM. OCs/TC-OCs were separately seeded onto glass plates and cultured with α-MEM (10% FBS) supplemented with a stimulus

(25 µg/l M-CSF and 50 µg/l RANKL) for 24 h. Nanomanipulation of calcified tissue-modified AFM probes (Supplementary Fig. 8) on OCs or TC-OCs was performed by writing nanolithography scripts to control the movement of the probe. As expected, the naive OCs could not firmly adhere to the probe owing to the poor adhesion force of only $0.05 \pm 0.01$ nN (Fig. 2h, j and Supplementary Movie 1). In contrast, the TC-engineered OCs

**Fig. 1 Encapsulating OCs with TC. a** Schematic of the structure of TC-OCs, and the working principle for their use as a bionic treatment for HO. **b** Representative images of TRAP staining ($n = 5$ independent experiments) of mature OCs after trypsin digestion. Bar: 300 μm. **c** Quantitative analysis of cell viability under different TC solution concentrations ($n = 5$ independent experiments). **d** Representative flow cytometry analysis of naive OCs and TC-OCs (160 μg/ml). **e** Flow cytometry analysis of OC cell surfaces positive for TC coating ($n = 3$ independent samples per group, ****$p < 0.0001$). Statistical comparisons were made with unpaired $t$-tests. **f** IR analyses of the TC reaction with OC membranes. **g** Fluorescence detection of OCs, TC-OCs (0 day), and TC-OCs (4 days) by CLSM, showing TC (green), cell membrane (PKH26, red), and the nucleus (Hoechst33258, blue, $n = 3$ independent cells per group). Bar, 20 μm. **h, i** Representative TEM micrograph of OCs/TC-OCs on the bovine cortical bone slice surface. Red asterisk, OCs; black arrow, resorption pit ($n = 3$ bone slices/group). Bar, 10 μm. Data are represented as mean ± SD. Source data are provided as a Source Data file (***$p < 0.001$, ****$p < 0.0001$).

firmly adhered to the probe without detachment (Fig. 2i, k and Supplementary Movie 2), as the adhesion force to the calcified tissue probe was increased by approximately twofold compared with that of the control ($0.14 \pm 0.019$ nN) (Fig. 2l, m), suggesting an enhanced adhesion force of TC-OCs to calcified soft tissues. Moreover, CLSM results further confirmed that the cells attached to the tipless cantilever were multinucleated (Fig. 2j, k). Finally, we performed additional experiments to measure the cell-migration capability and viability using live/dead probes (Fig. 2n, o). As depicted in Fig. 2p, approximately $67 \pm 3$ cells migrated into calcified tissue in the TC-OC group, but only $30 \pm 9$ cells migrated into calcified tissue in the native-cell group. Live/dead staining results suggested that compared with naive OCs, TC-OCs had increased adhesion to calcified tissue. This increase in the adhesion force confirmed the efficient targeting effect of TC-OCs to calcified soft tissues, indicating their potential for the biological decalcification of HO.

**Ectopic calcification resorption by engineered OCs (in vitro).** Calcium-mineral resorption is achieved by the release of protons from OCs[42]. Accordingly, the ability of TC-OCs to resorb calcified tissue can be analyzed using a pH-sensing chemical probe such as an AIE[TM] pH probe, which changes color from red to blue with increasing pH[43]. CLSM results showed that red signals were detected and multinucleated TC-OCs targeted to ectopic calcified tissue (Fig. 3a). To further prove the cells migrated to calcified tissue were mature osteoclasts, the calcified tissues were stained with TRAP, as the standard for osteoclast identification. The results further manifested that the cells that migrated to calcified tissue were TRAP-positive cells (see Supplementary Fig. 9a, b). More cells were observed on ectopic calcified tissue compared with no modification, indicating that cell adhesion and migration capacity was improved due to surface engineering (Supplementary Fig. 9c, d). Importantly, the fluorescence intensity of the red signal in the TC-OC group was substantially improved (approximately doubled) compared with that observed in the OC group, which indicated that engineered OCs have an increased ability to release acid relative to naive OCs (Fig. 3b). The overall appearance of the native OCs and engineered OCs (pink, Fig. 3c) that migrated into ectopic calcified tissue was analyzed by SEM. Consistent with the above results, more TC-engineered cells were observed on ectopic calcified tissue. Furthermore, an analysis of the acid-etched area by SEM further showed that the ability of TC-OCs to resorb calcium in the acid-etched ectopic calcified tissue areas was increased to approximately 2.5-fold (OCs: $21.9 \pm 9.4\%$; TC-OCs: $55.7 \pm 3.4\%$) (Fig. 3d–f).

Microcomputed tomography (micro-CT) reconstruction was performed on ectopic calcified tissue samples to assess decalcification at the histological level (Fig. 3g). The volume of calcification at 0 and 2 weeks was quantitatively estimated by micro-CT. The percentage of bone-volume reduction was used to evaluate the efficiency of bone resorption by engineered osteoclasts. The total ectopic calcification volumes in the OC

and TC-OC groups were reduced by $33.7 \pm 4.6\%$ and $67.0 \pm 1.6\%$, respectively (Fig. 3h), compared with that observed in the control group ($0.7 \pm 3.7\%$). Because of the increased cell migration, the amount of $Ca^{2+}$ eluted from the acid-etched areas estimated by ICP-OES was used to compare the bone-resorption capability between the native OCs and the modified cells. The amount of $Ca^{2+}$ eluted from the acid-etched areas into the culture medium in modified OC groups was increased by about three times (Fig. 3i) compared with native OC groups, verifying an improvement in ectopic calcified tissue resorption capacity after surface modification. All these in vitro results provide direct evidence of the advantages of TC-OCs in biological decalcification.

**Reversal of heterotopic ossification by engineered OCs in vivo.** Given the enhanced ectopic calcified tissue resorption capacity of TC-OCs, three separate models of HO were studied to observe the function of TC-OCs in vivo: (i) tenotomy, (ii) intramuscular model, and (iii) genetic model (Fig. 4a and Supplementary Fig. 10)[44]. The detailed methods of three separate HO models were provided in the supplementary files. Notably, the engineered OCs with TC molecules on the cell surface can be stored stably in vivo over four days (Supplementary Fig. 11). As depicted in Fig. 4b, we injected 0.9% NaCl (100 μl, normal saline, blank), OCs (100 μl, $10^6$ cells/ml/500 g, control), or TC-OCs (100 μl, $10^6$ cells/ml/500 g) in situ at the Achilles tendon calcification site every four days. Localization to the calcified site was guided by living micro-CT and accomplished with an in situ injection. In the tenotomy model, HO maintained growth in the blank group but showed a volume reduction in the control (OC) group and the TC-OC group based on two- and three-dimensional micro-CT at 30 days after injection (Fig. 4b and Supplementary Fig. 12). Consistent with the micro-CT images, tenotomized rats treated with TC-OCs exhibited reduced HO volumes at 30 days (by approximately $31.8 \pm 10.2\%$) compared with the initial volume (at 0 day) (Fig. 4c). However, in the OC groups, HO volumes were reduced at 30 days (by approximately $18.6 \pm 11.2\%$) compared to the initial volume (at 0 day). It should be noted that there was an increase in bone mass in the blank groups ($-19.6 \pm 19.1\%$). TC-OC treatment resulted in the absorption of most of the ectopic bone after 60 days, as shown by analysis and quantitative comparison (Fig. 4c, d). Quantification of the relative BV showed that ectopic bone formation was reduced by more than $70.5 \pm 7.6\%$, compared with the $33.3 \pm 7.9\%$ observed in the OC group after 60 days, which indicates that engineered OCs exhibit superior ability to resorb HO. We next confirmed these findings in the intramuscular model, which demonstrated a reduction in total HO volume at 30 days (Fig. 4e, f). In addition, we noted that soft tissue HO was nearly completely abolished in TC-OC treated group at 60 days, consistent with our findings in tenotomy model.

To strengthen our findings, we next used $Mkx^{-/-}$ knockout mouse that formed HO after being born at eight weeks as genetic model. Again, our concepts were proved using TC-OCs in treated mice, which showed less ectopic bone at 30 days, as shown by

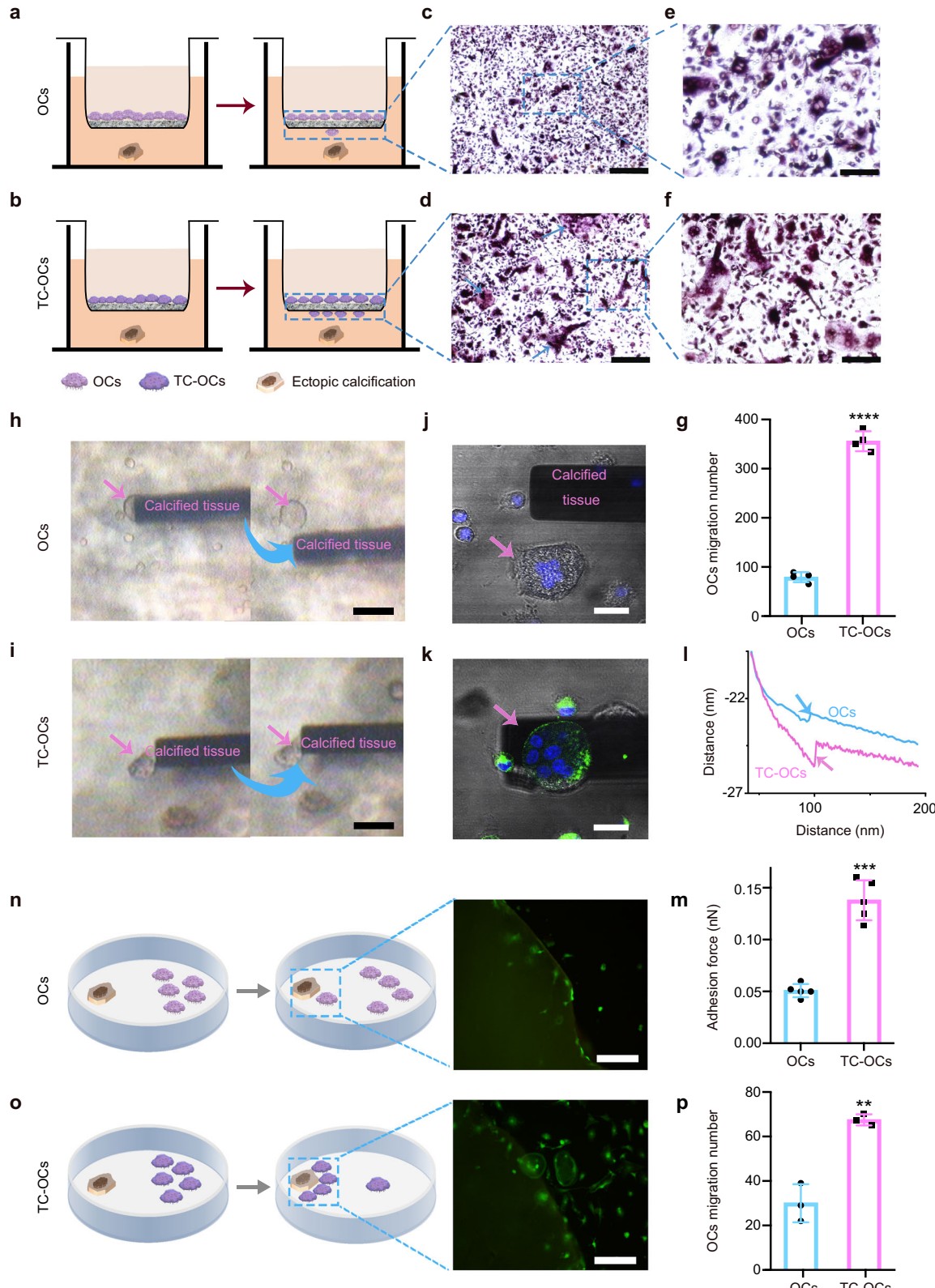

micro-CT and quantitative comparison (Fig. 4g, h and Supplementary Fig. 12). After 60 days, TC-OC group developed minimal HO around the calcaneus, and these lesions were substantially smaller than in OCs and blank groups. Taken together, the results were striking and proved the substantially improved efficacy of TC-OCs to reversibly resorb ectopic bone over naive OCs in a different type of HO model.

Bone-resorption capacity of engineered TC-OCs in other calcified tissue, cells were further proved in endochondral bone (cranial bone) and intramembranous bone (tibia bone) in vitro. As depicted in Supplementary Fig. 13 and Supplementary Fig. 14, BMD, BS/TV, and BV/TV were decreased in TC-OC groups compared with OC-treated groups, confirming that TC-engineered OCs promote excellent ability of bone resorption in

**Fig. 2 Improved adhesion force between TC-OCs and calcified tissue. a**, **b** Schematic depicting the coculture assay with a conventional Transwell design for the detection of OC/TC-OC migration to ectopic calcified tissue. **c**, **d** Typical images showing migrated OCs/TC-OCs after 12 h of incubation ($n = 3$ independent samples/group). Bar, 200 μm. **e**, **f** Enlargement of **c**, **d**. Bar, 100 μm. **g** Quantitative analysis of migratory OCs/TC-OCs that were manually analyzed using Image J ($n = 3$ independent samples/group, ****$p < 0.0001$). **h**, **i** Representative images of OCs/TC-OCs with probes attached to calcified tissues ($n = 5$ independent samples/group). Bar, 100 μm. **j**, **k** Representative CLSM images of the probe after treatment with OCs/TC-OCs ($n = 3$ independent samples/group, tetracycline (green), nucleus (Hoechst 33258, blue)). Scale bar, 20 μm. **l** A representative force curve of OCs/TC-OCs with calcified tissue. **m** Quantitative analysis of the force between OC/TC-OC cell membranes and probes attached to calcified tissue. Force curves were calculated from the frequency-shift difference curves using the Sader–Jarvis method ($n = 5$ independent samples/group, ***$p = 0.0006$). **n, o** Schematic of the test design for the detection of OC/TC-OC migration to ectopic calcified tissue. Live/dead staining images of OCs/TC-OCs migrating to calcified tissue. Bar, 300 μm. **p** Quantitative analysis of migratory OCs/TC-OCs ($n = 3$ independent samples/group, **$p = 0.0019$). Data are represented as mean ± SD. Statistical comparisons were made using either unpaired (**g**, **p**) or paired (**m**) $t$-tests. Source data are provided as a Source Data file (**$p < 0.01$, ***$p < 0.001$, ****$p < 0.0001$).

---

endochondral and intramembranous bone. These data support the notion that surface-modified osteoclasts have the potential not only for HO-reversing treatment but also for the treatment of other types of ectopic calcification.

Pathological analyses of tendon slices from each group included Von Kossa (VK), hematoxylin and eosin (H&E) staining, Masson's trichrome (Masson), TRAP staining, and Alcian blue(AB) staining. The VK and H&E staining results suggested that ectopic tissues were prominent in the blank and control groups, but clearly reduced in rats treated with TC-OCs (Fig. 4i). Consistent with these results, the Masson and AB staining results showed that blank and control rats exhibited abundant endochondral bone and cartilage and that oriented collagen fibers were damaged. Furthermore, the ectopic tissues in rats were efficiently diminished and treated with TC-OCs. TRAP staining images and quantitative statistics demonstrated that more TRAP-positive cells were maintained and fewer ectopic calcifications were retained in TC-OC-treated rats than in the other rats. These pathological results demonstrated the effective and precise bone resorption of TC-OCs in the setting of HO.

To further examine the ability of TC-OCs to target and adhere to ectopic calcification in vivo compared with OCs, cells were in situ injected into the Achilles tendons of the model rats[44], and these tendons were dissected after 0 h and 2 h. For the specific method of tendon injury used to induce HO, refer to the supplementary file. As shown in Supplementary Fig. 15a, b, and c at high spatial distribution, deep tissues were visualized at various depths with a view up to 100-μm deep by two-photon CLSM. The location and shape of the ectopic calcifications and the density of OCs/TC-OCs in each Achilles tendon were clearly detected under two-photon CLSM at 0 h in the three groups. CLSM images of the OCs/TC-OC distribution in the calcification in tendon were captured at 2 h after injection (Supplementary Fig. 15d, e and f). The number of TC-OCs at the calcification sites was increased compared with that in the OC group (Supplementary Fig. 15g), implying that the engineered OCs have an adhesion and calcification-targeting ability superior to that of naive OCs. Importantly, our results demonstrated that the injected TC-OCs around the HO site were bioactive during the treatment. In summary, in an Achilles' tenotomy rat model, the injection of TC-OCs around the calcified tendons in situ enabled the visualization of the local sites of calcified mineral resorption following reversible recovery from HO.

To qualitatively and quantitatively evaluate the cell viability of osteoclasts and TC-engineered osteoclasts in vivo, OCs with or without surface modification were used to perform in situ injections for an extended time period. We visualized the cell viability of native and engineered OCs labeled by Cell Trace Far Red DDAO-SE fluorescent tag in vivo based on the in vivo living imaging at 0 day and 4 days (Supplementary Fig. 16). The results showed that

labeled TC-OCs showed a stronger staining intensity than OCs. The long-term in vivo circulation of the engineered OCs in ectopic tissue demonstrated that the TC-OCs retain good cell viability and function for more than four days. Moreover, we quantified the cell viability of native and engineered OCs in vivo by flow cytometry at 0 day and four days after injection. Flow cytometry further revealed increased fluorescence signals of Calcein AM in the TC-OC groups (Supplementary Fig. 17a, b, c), and the values were improved from $27.6 \pm 7.7\%$ in OC groups to $76.6 \pm 10.0\%$ in TC-OC groups. According to the improvement of cell viability with the modification of the cell membrane (Supplementary Fig. 17d), we deduced that the TC-engineered OCs increase their unique function of bone resorption. The results demonstrated that surface engineering is essential for natural OCs to maintain the key functions of bone resorption in vivo for long-term circulation. Long-term circulation in vivo further demonstrated the favorable bone-resorption activity of TC-engineered OCs and the feasibility of HO treatment.

The biocompatibility and activity of the engineered OCs in vivo was further proven by testing cytokine response in tendon lysis. The cytokine response of bone formation and resorption in the area of the cell implants was measured by ELISA (alkaline phosphatase (ALP), bone morphogenetic protein 2 (BMP-2), TRAP, and type-I collagen cross-linked C-telopeptides (CTX)). Although the expression of proteins associated with bone formation (ALP and BMP-2) in the area of the cell implant is lower than that in the control groups (Supplementary Fig. 18), proteins related to bone resorption (CTX and TRAP) were more highly expressed in the TC-OC groups. These results indicated that surface engineering leads to slight reduction of bone-formation marker and presents good biocompatibility and bone-resorption activity. The expression of proteins associated with bone resorption had improved, thus confirming the conceptual potential of this approach to HO-reversing treatment.

At the protein level, the key factors involved in cell differentiation to the HO osteogenic lineage are bone morphogenetic proteins (BMPs) and inflammatory factors[45]. Blood-based detection revealed no changes in the levels of ALP, BMP-2, interleukin 6 (IL-6), or tumor necrosis factor alpha (TNF-α) in the TC-OC group compared with the control group (0.9% NaCl), indicating that no inflammatory effect and side effects were induced by the "living treatment agents" (Supplementary Fig. 19).

Bone resorption by osteoclasts is normally coupled to bone formation by osteoblasts[46]. TRAP and CTX were used as standard biomarkers for bone resorption. The serum levels of TRAP and CTX were normal (Supplementary Fig. 20), suggesting that surface engineering did not lead to side effect. Furthermore, no obvious changes were observed in blood chemistry or bone density (Supplementary Fig. 21 and Supplementary Fig. 22) or in trabecular morphology, confirming the biocompatibility nature of TC-OCs in the treatment of HO.

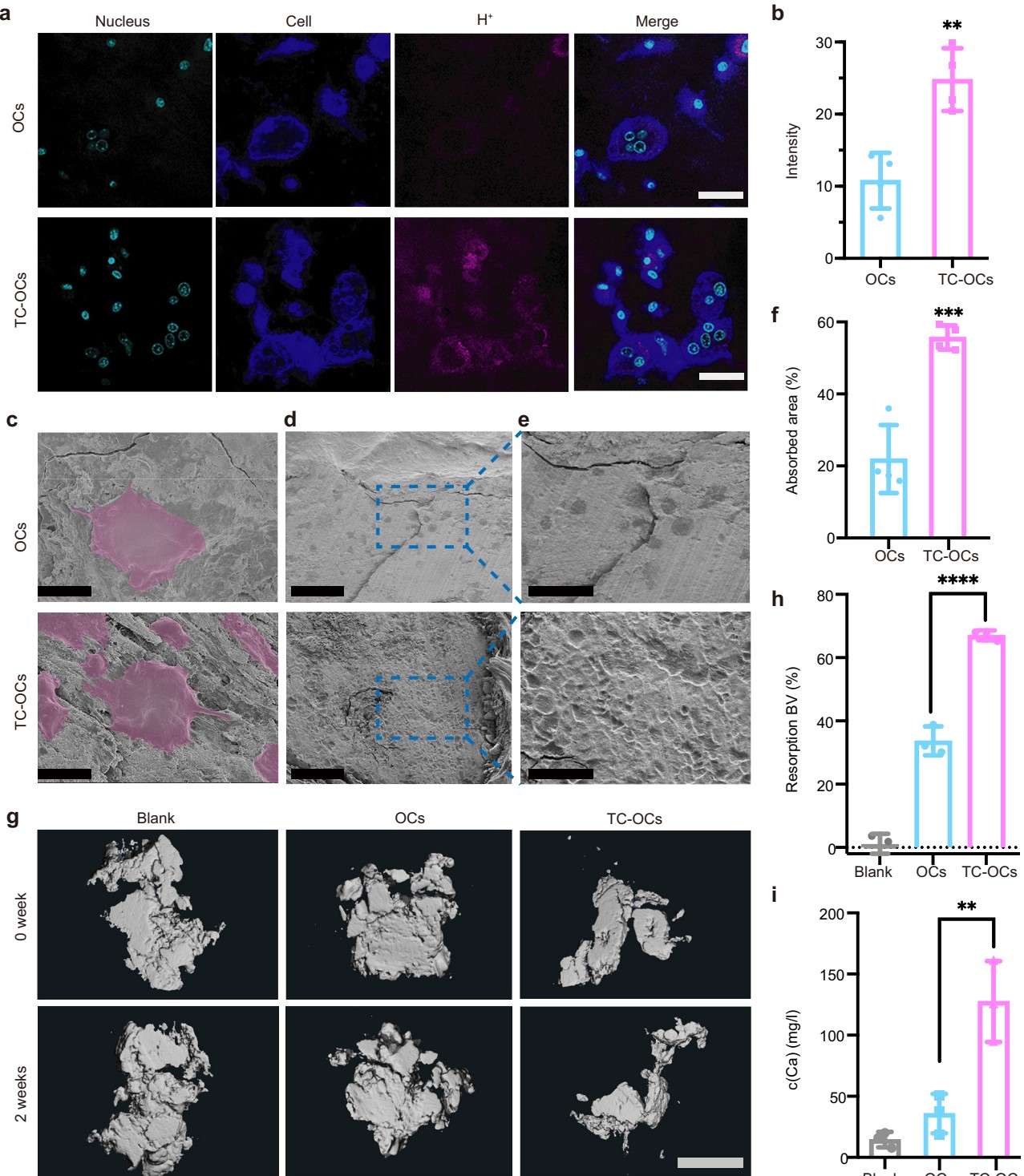

**Fig. 3 Increased ectopic calcified tissue resorption capacity of TC-OCs in vitro. a** CLSM images of OCs/TC-OCs seeded into 48-well plates containing ectopic calcifications and incubated with an AIE$^{TM}$ pH probe. Scale bar: 20 μm. **b** Quantitative intensities of OCs/TC-OCs labeled with the AIE$^{TM}$ pH probe ($n = 4$ independent samples/group, **$p = 0.003$). **c** SEM images of OCs and TC-OCs migrating into calcified tissue in vitro ($n = 4$ independent samples/group). Scale bar: 30 μm. **d** SEM images of calcified tissue resorption area ($n = 4$ independent samples/group). Scale bar: 300 μm. **e** Enlarge of (**d**). Scale bar: 100 μm. **f** Statistical analysis shows the changes in the resorption area between the two groups ($n = 4$ independent samples/group, ***$p = 0.0005$). **g** Three-dimensional reconstruction of micro-CT scans of ectopic calcified tissue treated with 0.9% NaCl (blank), OCs, or TC-OCs at 0 and 2 weeks ($n = 3$ independent samples/group). Scale bar: 2 mm. **h** Resorption bone-volume (BV) analysis showing the changes in resorption areas among the three groups ($n = 3$ independent samples/group, ****$p < 0.0001$). **i** Ca$^{2+}$ eluted from the acid-etched areas in three groups, as measured by ICP-OES ($n = 3$ independent samples/group, **$p = 0.0015$). Data are represented as mean ± SD. Statistical comparisons were made by using unpaired (**b, f**) $t$-tests and ordinary one-way analysis of variance (ANOVA) with multiple-comparison tests (**h, i**). Source data are provided as a Source Data file (**$p < 0.01$, ***$p < 0.001$, ****$p < 0.0001$).

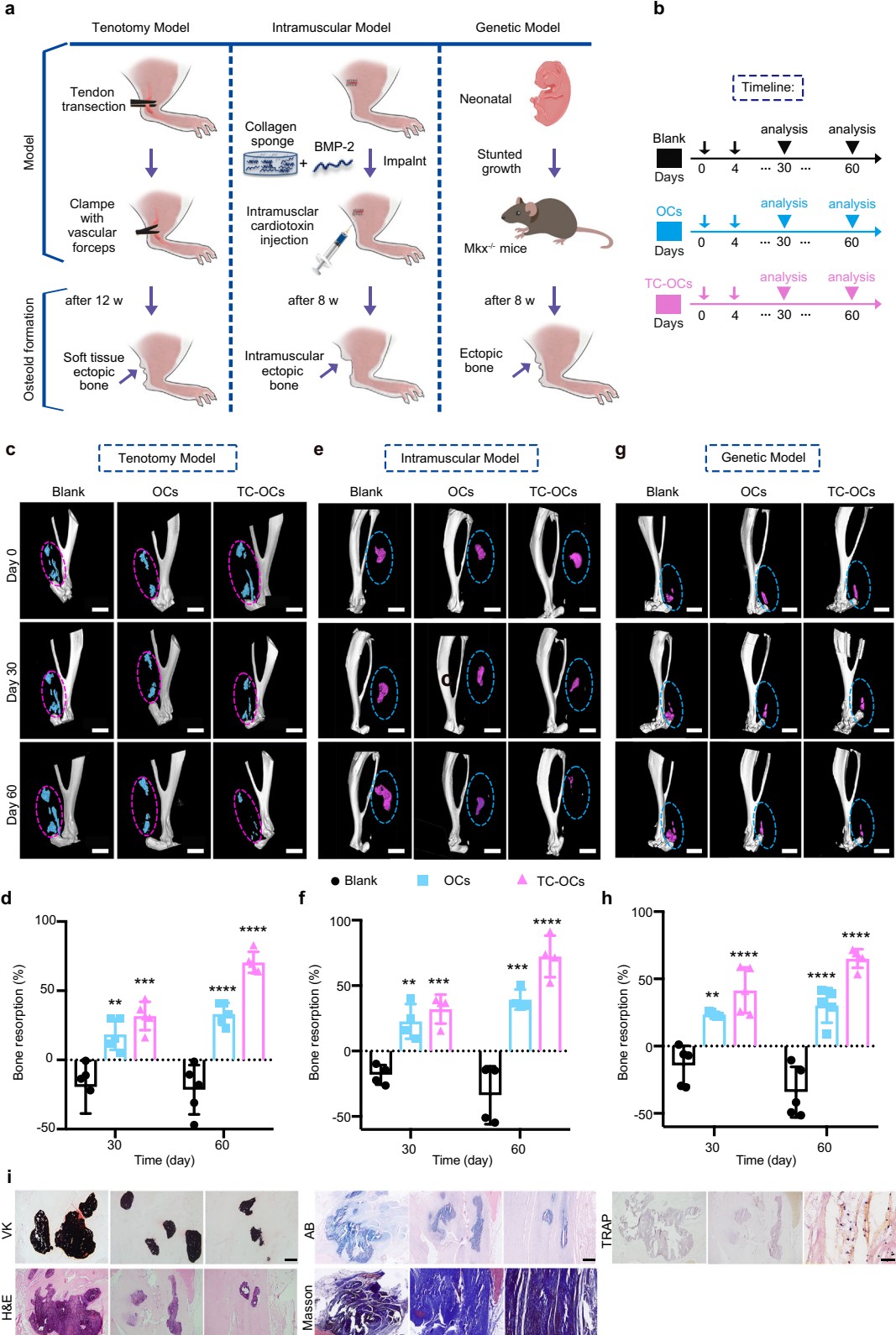

There may be concerns regarding the biosecurity of TC. TC has been widely used as an antibiotic and for bone-targeting purposes[36,47]. In our experiment, 100 μl of TC-OC solution (160 μg/ml) was used in each injection, and the total amount of TC was only 2 μg, i.e., far less than the threshold levels (100 mg/kg) set by the Steering Committee of the Veterinary International Committee on Harmonization.

Another potential concern regards the phenomenon referred to as the "rebound effect" often observed in HO treatment[11]. Ectopic calcification tissues grew in control rats; however, only minimal tissue growth was observed in rats treated with TC-OCs, indicating the lack of a rebound effect (Supplementary Fig. 23).

**Fig. 4 Increased bone resorption capacity of TC-OCs for three types of HO in vivo. a** Three different types of HO animal models: tenotomy model, intramuscular model, and genetic model. **b** Timeline of the injected OCs/TC-OCs. **c, d** Three-dimensional reconstructions of micro-CT scans and bone resorption of tenotomy model rats treated with OCs, TC-OCs or 0.9% NaCl (blank) (for 0, 30, or 60 days). Large mineralized HO masses were visible by micro-CT in rats ($n = 5$ animals per group). Scale bar: 3 mm. Resorption ratio of day 30 **$p = 0.0028$, ***$p = 0.0002$. Resorption ratio of day 60 ****$p < 0.0001$, ****$p < 0.0001$. **e, f** Micro-CT scans and bone-resorption ratio of intramuscular model ($n = 4$ animals). Scale bar: 3 mm. Resorption ratio of day 30 **$p = 0.0012$, ***$p = 0.0003$. Resorption ratio of day 60 ***$p = 0.0004$, ****$p < 0.0001$. **g, h** Micro-CT scans of genetic model ($n = 5$ animals per group). Scale bar: 2 mm. Resorption of genetic model HO was evaluated by measuring the BV resorption ratio of ectopic masses in the blank versus the OCs and TC-OC groups on days 0, 30, and 60. Resorption ratio of day 30: **$p = 0.0015$, ****$p < 0.0001$; Resorption ratio of day 60: ****$p < 0.0001$, ****$p < 0.0001$. **i** Ectopic tissues were sectioned and examined by VK, H&E, AB, Masson, and TRAP staining; bar: 500 μm. Data are represented as mean ± SD, all statistical comparisons were made by using an ordinary one-way analysis of variance (ANOVA) with multiple-comparison tests. Source data are provided as a Source Data file (**$p < 0.01$; ***$p < 0.001$; ****$p < 0.0001$).

## Discussion

More than one million patients worldwide undergo joint-replacement surgery each year, and 20% of these patients will develop HO[48]. In addition, heterotopic bone often forms in burn and trauma patients[49]. According to the researchers, more than 80% of patients will develop HO after joint-revision surgery, which represents an enormous medical and financial burden. Furthermore, over approximately 20% of patients who experience craniocerebral trauma and undergo spinal surgery may develop HO, and HO has even been reported to form outside the skeleton in regions of chronic inflammation, including the abdominal wall and amputation sites[50,51]. Despite the high incidence of HO associated with commonly performed operations, our deep understanding of the key pathway that causes different types of HO, and the availability of preventive treatment strategies, current biotechnological methods cannot reverse HO once a calcified site has formed[49]. However, the results of our current proof-of-concept study show that rational cell engineering of OCs can support HO treatment strategies. An exciting and important in vivo result of this study is that HO could be effectively eliminated without a "rebound effect". Furthermore, engineered OCs have potential for use in the treatment of different type of ectopic calcification diseases.

We should emphasize that this cell-based treatment of HO has been tested in only a conceptual study, and a great deal of research is needed before it can be developed for clinical use. Based on our current understanding, several issues should be appropriately addressed to ensure the success of future translational studies. First, immune cells in the bone marrow interact with osteoclasts and osteoblasts through cell-surface molecules, thereby regulating bone homeostasis[52,53]. Because OCs can ultimately promote the differentiation of osteoprogenitors, the longer-term limitation for the therapy is that the presence of OCs within the wound may encourage localized ossification and weaken the role of engineered OCs. In addition, a healthy balance between bone formation by osteoblasts and bone resorption by osteoclasts maintains bone homeostasis[46]. Despite the vigorous regulation and control of bone equilibrium, changes in remodeling can occur as a result of imbalances between osteoclast and osteoblast activity, which results in debilitating bone diseases such as osteoporosis and HO[8,54]. As a modified material in our system, TC is easily deposited in minerals of the human body and can be a potent chemical compound for bone targeting that increases cell adhesion to calcified tissue so as to promote cell activity for HO decalcification[33]. However, in clinical applications, most TC is ineffective and cannot be removed by conventional methods, and this situation has become even worse due to the misuse and abuse of TC[55]. Accordingly, it is essential to develop modified materials with greater biosafety and that are more environmental friendly for subsequent application studies. Finally, additional investigations to evaluate the systemic bioavailability and the long-term biosafety of the engineered cells are

of crucial importance. Although we performed the current proof-of-concept study to assess the treatment of HO by OCs, more specific experimental verification for clinical applications should be performed.

Natural organisms have the advanced function of being artificially irreproducible, and our investigation followed a proof-of-concept approach of "living treatment agents", which can further improve biological cells by chemical modification. In this study, as a proof of concept, we proposed that the chemically improving OCs can extend their functions for HO treatment. We are keen to see advancements in other calcification diseases, such as vascular calcification, facilitated by the use of material-based OC encapsulation approaches. Such an incorporation of functional materials into cells shows promise for cellular engineering by combining biology, chemistry, and materials sciences to design artificially improved osteoclast as "living agents" for biomedical applications to HO-reversible treatment.

## Methods

**OC differentiation.** Six-to-eight-week-old male mice (C57BL/6, Shanghai SLAC Laboratory Animal Co., Ltd., China) were sacrificed by decapitation under deep anesthesia with 3% pentobarbital sodium. Bone marrow cells (BMMs) were isolated from the tibiae and femurs and cultured in α-MEM (Gibco, USA) containing 25 μg/l M-CSF (R&D system), 10% FBS (Gibco, USA), 100 U/ml penicillin G, and 100 lg/ml streptomycin for 24 h at 37 °C overnight under 5% $CO_2$. BMMs were seeded onto 24-well plates ($4 \times 10^4$ cells/well) or 96-well plates ($1 \times 10^4$ cells/well) and cultured for six days in α-MEM supplemented with 10% (v/v) FBS, 25 μg/l M-CSF, and 50 μg/l RANKL (R&D system) at 37 °C with 5% $CO_2$. The culture medium was replaced with fresh medium every other day.

**OC chemical modification.** Mature OCs were digested with 0.25% EDTA-trypsin for 3–5 min and then collected by centrifugation ($100 \times g$, 10 min; Centrifuge 5418, Eppendorf, Germany) and washed three times with PBS ($1 \times$ PBS, 10 mM, pH 7.2). Typical engineering conditions involved 1 ml of cell suspension ($2 \times 10^5$ cells/ml) mixed with tetracycline hydrochloride (Aladdin reagent, AR) to prepare 1 mg/ml ($V = 160$ μl) solutions, with additional shaking of the centrifuge tube for 1 h. The final TC concentration is 160 μg/ml.

**Cell-surface characterization.** OCs were engineered through the chemical modification methods described above. Then, the cells were stained with the fluorescent cell linkers PKH26 (250 nM, red, Sigma-Aldrich, USA) and Hoechst 33258 (5 μM, blue, Beyotime, China) for general cell membrane and nucleus labeling. The cells were then rinsed with PBS and seeded in 3.5-cm wells with coverslips before being subjected to confocal laser-scanning microscopy (CLSM). Control cells were stained with 250 nM PKH26 for general cell-membrane labeling for 3 min and with 5 μM Hoechst 33258 for another 3 min, with no TC treatment. Then, the cells were rinsed with PBS three times and subjected to CLSM (FV1000, Olympus) at 0 and 4 days. Subsequently, native OCs and TC-OCs were cultured with α-MEM (10% FBS) and then continuously subjected to CLSM (Nikon) at 1, 2, and 3 days. All images were captured and analyzed with image analysis software (Olympus FluoView FV1000 version 2.1b). To study the interaction between TC molecules and the cell membrane, OCs/TC-OC (50 μl, $1 \times 10^6$/ml) cell suspensions were placed between two sealed KBr plates for the measurements. FTIR spectra were measured by an IR Affinity-1 infrared spectrometer (Shimadzu, Japan) with 32 scans at a resolution of 2 $cm^{-1}$. The backgrounds were determined using a blank, which was 0.9% NaCl (50 μl). Raman spectra (InVia Reflex, Renishaw, England) were used to analyze the surface characterization of TC-OCs. The excitation wavelength was 532 $cm^{-1}$, the wavenumber is 1000–2000 $cm^{-1}$.

Fluorescence spectrophotometry (RF-5301, Shimadzu, Tokyo, Japan) analysis of OCs and TC-OCs (300 μl) was performed at an excitation wavelength of 450 nm.

**Flow cytometry analysis**. To evaluate encapsulation efficiency, surface-engineered OC was stained with 250 nM PKH26, which is not membrane penetrable. Cell-surface engineering efficiency was indicated by TC (160 μg/ml) autofluorescent on cell membranes. Mature OCs were digested by trypsin with 0.25% EDTA for about 20 mins and washed with PBS three times. Then, OCs were encapsulated at different concentrations according to the surface-engineering method described above. Control cells were stained with 250 nM PKH26 for general cell-membrane labeling for 3 min and with 5 μM Hoechst 33258 for another 3 min, with no TC treatment. The stained cells were washed, collected, and adjusted to 300 μl of cell-suspension solution ($1 \times 10^6$ cells/ml), and then analyzed on a CytoFLEX LX (Beckman Coulter, US, flow-cell dimensions: 430-μm × 80-μm internal diameter). The fluorescence of naive cells was used as a control and cells were filtered through 200-mesh cell filters. The linear gains were set in the light scatter channel, and the fluorescence channel was set on a logarithmic scale. In total, 10,000 cells were analyzed in each condition. Every sample was conducted to three replicates and protected from light. Finally, the fluorescence intensity was measured by FlowJo software (FlowJo vX.0.7).

**Bone and calcified tissue resorption**. To verify whether the bone-resorption capacity of OCs is affected after TC engineering, bovine cortical bone slices (Shanghai Lushen Biotechnology Company, ø6 mm) were prepared in a sterile environment and placed into the bottom of 96-well plates. OCs/TC-OCs ($1 \times 10^4$ cells/well) were seeded onto bovine cortical bone slices and then cultured in α-MEM (10% FBS) with a stimulus (25 μg/l M-CSF and 50 μg/l RANKL) as described above for six days. After that, the slices were washed with ammonia, and images were captured with a scanning electron microscope (SEM) (SU-8010 Hitachi Co, Tokyo, Japan). The absorbed areas were calculated by ImageJ software (version 1.6.0).

To assess calcified tissue resorption in vitro, eight-week-old male Sprague-Dawley rats (body weight 290–330 g) were preformed tenotomy-induced HO. All animals were housed under a 12-h light/dark cycle at controlled room temperature of 22–24 °C and a relative humidity of 40–70%. Rats were maintained at Sir Run Run Shaw Hospital, Zhejiang University School of Medicine. All animal experiments used in this study were approved by the Institutional Animal Ethics Committee of Sir Run Run Shaw Hospital of Zhejiang University (Number: SRRSH2021401 and 201801224). Specific animal model methods were listed in the Animal Model section in the supplementary file. After routine feeding and observation for 12 weeks, the animals were subjected to X-ray analysis (Faxitron MX-20, USA) and micro-CT (Suzhou Hiscan Information Technology Co., Ltd.). Quantitative analysis of calcified tissue BV by micro-CT. Nine calcified tissue samples were randomly divided into three groups: blank (0.9% NaCl), OCs, and TC-OCs. The BV values were $28.08 \pm 6.01$ mm$^3$, $31.42 \pm 9.21$ mm$^3$, and $28.55 \pm 3.65$ mm$^3$ before cell treatment, respectively. Then, OCs or TC-OCs ($2 \times 10^4$ per well) were seeded onto one side of a well in a 48-well plate, and ectopic calcified tissue was seeded on the other side of the well. Then, cells were seeded every two days and cultured in α-MEM (10% FBS) with 25 μg/l M-CSF and 50 μg/l RANKL. The BV of calcified tissues after TC-OC treatment was obtained by micro-CT in two weeks. The BV resorption rate of calcified tissue is calculated from the BV of calcified tissue before OC treatment (0 weeks) and the BV after treatment (2 weeks). Formula (1) is

$$\frac{\text{BV}(0\,\text{week}) - \text{BV}(2\,\text{week})}{\text{BV}(0\,\text{week})} \times 100\% \tag{1}$$

For quantitative analysis of the bone-resorption capacity of OCs/TC-OCs in vitro, $Ca^{2+}$ eluted from the acid-etched areas of culture media in 48-well plates (1 ml of 10% FBS α-MEM) was analyzed by inductively coupled plasma optical emission spectrometry (ICP-OES) (Thermo iCAP 10 6300, USA).

The medium was diluted to 10 ml, and an analytical calcium-standard solution (Aladdin Company) of $100.00 \pm 0.10$ μg/ml in 5% nitric acid was used as a stock solution for calibration. All solutions were filtered through 0.22-μm Millipore films prior to use. The concentration of $Ca^{2+}$ eluted from the acid-etched areas of culture media was obtained through those data. Formula (2) is

$$C_{Ca}(\text{elution}) = C_{Ca}(2\,\text{week}) - C_{Ca}(0\,\text{week}) \tag{2}$$

The amount of $Ca^{2+}$ eluted from the acid-etched areas into the culture medium was determined by ICP-OES, and the values were $35.85 \pm 16.10$ mg/l and $127.54 \pm 33.07$ mg/l.

**Transwell migration assay**. OCs/TC-OCs were digested with 0.25% trypsin-EDTA and centrifuged at $100 \times g$ for 10 min. The supernatant was aspirated, and the pellet was washed twice with PBS and resuspended in 100 μl of 1% FBS α-MEM medium. Approximately $3 \times 10^4$ cells were added to the upper chambers of the Transwell inserts (12-μm pores, Corning), which were positioned in the wells of a culture plate and placed in direct contact with 600 μl of (10% FBS) α-MEM with ectopic calcification. A 1-mm-thick ectopic calcification disk was prepared by making two parallel cuts perpendicular to the tissue axis using a slow-speed diamond saw (Isomet 1000, Buehler Ltd.). The plate was incubated for 12 h at 37 °C in

an incubator containing 5% $CO_2$. Then, the inserts were removed, washed twice with PBS, and gently wiped with a cotton swab moistened with PBS. The cells in the upper chamber were removed, and the cells in the lower chamber were fixed with 4% paraformaldehyde for 30 min, washed twice with PBS, and stained with TRAP for 1 h. After a final PBS wash, randomly selected fields on the membrane were observed under 10× magnification, and the number of cells was quantified using Image-J (version 1.6.0) software.

**Atomic force microscopy (AFM) experiment**. AFM was used to measure the force at the atomic level. To prepare for AFM measurements, OCs/TC-OCs were seeded at a density of $1 \times 10^4$ cells/well on a 12-mm microscope cover glass (Fisher Scientific,12-545-80) prepared in 48-well plates and then incubated for 24 h before imaging. One calcified tissue powder was attached to a tipless cantilever (CSG11/tipless, NT-MDT, force constant of 0.1 N/m as measured by the thermal tuning method). Both cells and cantilevers were thermally equilibrated at 37 °C for 40–60 mins prior to imaging to minimize thermal drift. For in situ AFM experiments, the cover glass containing cells was placed under the AFM liquid cell and modulated. Then, phenol red-free α-MEM (Gibco, USA, 10% FBS) supplemented with 25 μg/l M-CSF and 50 μg/l RANKL was injected into the liquid cell. The experiments were conducted on a commercial AFM (Nanoscope IVa, Veeco).

**HO treatment in vivo**. All materials were sterilized by ultraviolet radiation for 24 h. Before the experiment in vivo, the localized calcified site was guided by live micro-CT (Suzhou Hiscan Information Technology Co., Ltd.) and accomplished with in situ injection. It follows that TC-OCs were injected into the HO site directly. Every calcified tendon was treated every four days with 100 μl of 0.9% NaCl (normal saline, blank), OCs ($10^6$ cells/ml/500 g), or TC-OCs ($10^6$ cells/ml/500 g) in situ. To reduce the interference caused by the syringe needle, insulin syringes were used to inject materials into the Achilles tendon in situ. All rats were sacrificed 60 days after the initial injection, and their Achilles tendons were harvested and used for histological observations.

**Histological examinations**. The calcified tissue was decalcified in 10% EDTA and embedded in paraffin. A continuous series of 5-μm-thick sections was stained with H&E, Masson's trichrome, Von Kossa, Alcian blue, or TRAP at pH 1.0. Alcian blue-stained sections were counterstained with eosin.

**Micro-CT analysis**. Achilles tendons in living rats were analyzed in situ by Hiscan XM Micro CT (Suzhou Hiscan Information Technology Co., Ltd.) every 30 days. The X-ray tube settings were 60 kV and 133 μA. Images were acquired at a 9-μm resolution, and a $0.5^0$ rotation step through a 360° angular range with a 50-ms exposure per step was used. The scan resolution was 25 μm. The images were reconstructed with Hiscan Reconstruct software (version 1.0, Suzhou Hiscan Information Technology Co., Ltd.) and assessed by Hiscan Analyser software (version 1.0, Suzhou Hiscan Information Technology Co., Ltd.). Relative bone-volume (BV) ratios were calculated by dividing the experimental group's BV by the blank group's BV and used to determine differences between experimental and control values. The bone-mineral density (BMD), BV (bone volume) /TV (total volume), trabecular number (Tb. N), trabecular thickness (Tb. Th), and trabecular separation (Tb. Sp) data of normal bone were also assessed. In the intramuscular and genetic model, the ectopic calcification was analyzed by U-CT (MiLabs, Netherlands) and IMALYTICS preclinical (Version 2.1.8.9), and the scan resolution was 25 μm.

**Statistics and reproducibility**. The statistical significance of all experiments was determined by Student's $t$-test and one-way ANOVA is presented as the mean ± standard deviation (SD) of at least three samples from independent analyses, as indicated. Statistical tests were performed by GraphPad Prism 8 for macOS (Version 8 (131), USA). $P$ values were considered as statistically significant as follows: no significance (ns) $p > 0.05$, *$p < 0.05$, **$p < 0.01$, ***$p < 0.001$, ****$p < 0.0001$. All experimental findings were replicated independently and reproducible with three times or more.

**Reporting summary**. Further information on research design is available in the Nature Research Reporting Summary linked to this article.

## Data availability
All data generated in this study are provided in the main text, Supplementary Information and Source Data files or are available from the corresponding author upon reasonable request. Raw data of this study have been deposited at Zenodo database with open access (https://zenodo.org/record/5602506#.YXj2JBBBzLZ). Source data are provided with this paper.

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

## Acknowledgements

We thank Xiaoyu Wang, and Biao Jin for constructive discussions. We thank Shuang-shuang Liu and Wei Yin for the assistance with CLSM data analysis, Jingyao Chen for their help with histological analysis, and Yanhong Chen for assisting with the animal model. This work was supported by The National Natural Science Foundation of China (21625105) and National Key R&D Program (2018YFC1105103).

## Author contributions

**Conceptualization:** R.T. and W.J. initiated this study. Z.X. and R.T. supervised and supported the research. W.J. performed most experiments, including OC induction, TC engineering, as well as characterizations. W.J., X.L., C.Z., P.Q., H.W., X.C. and H.O.

performed the Achilles' tenotomy model, intramuscular model and the animal experiments. W.J., Y.Z., R.Z. and Z.H. performed flow cytometry of TC-OC cell viability in vivo. W.J., H.P., Z.X. and R.T. analyzed the data. The paper was written by W.J. and R.T. All authors provided critical feedback and helped shape the paper.

## Competing interests
The authors declare no competing interests.

## Additional information

**Peer review information** *Nature Communications* thanks Cory Xian and the other anonymous reviewer(s) for their contribution to the peer review this work. Peer reviewer reports are available.

