## [Peer Review File · Nature Communications]

Reviewers' Comments:

Reviewer #1:

Remarks to the Author:

The manuscript by Jin et al examines bone marrow-derived cells coated with tetracyclin for their ability to attach to and resorb heterotopic ossification in vitro and in vivo. There are several challenges with this manuscript.

As written, the first sentence in the abstract is incorrect: "their unique decalcification activity is strictly limited to the removal of ectopic calcification". It's clear from the remainder of the abstract that the authors do not mean to state that osteoclasts can only remove ectopic calcification, which is what is stated. Please revise.

It's clear that this manuscript needs to be edited by someone with knowledge of osteoclast biology with a better command of the English language. There are many sentences that are not correct. For example: lines 11-12 of the Introduction: "OCs are multinucleated cells induced by the differentiation of monocyte/macrophage cells at or near bone." Osteoclasts are not "induced by the differentiation". They are induced by RANKL and M-CSF produced by osteoblastic lineage cells that induce the differentiation of marrow-derived precursors. The next sentence begins with "As natural intelligent materials, OCs can remove large amounts of calcium,...". Osteoclasts are not "intelligent" nor are they "materials".

Specific concerns:

Methods:

Osteoclast cultures. This is unnecessarily complex. There is no need to discuss the dose/response studies, since it is so well-documented that M-CSF and RANKL are required for osteoclast differentiation.

Please define CLSM the first time it is used in the methods section. Is there a reason that two different sources are used for this? Please explain.

Quantitating TC molecules on the osteoclasts: This is confusing. Are you making the assumption that all of the TC molecules in the lower concentrations of your standard curve attach to the cells? If so, please make this assumption clear or explain your quantitation more clearly.

Flow cytometry: osteoclasts are large cells and most are sheared if the aperture of the machine is not sufficiently large. Please provide more precise details of the flow protocol. Were the cells evaluated visually to determine how intact there were after flow?

Ectopic resorption: What is the stimulus mentioned in this section? This should be clearly stated in the Methods section.

AFM What is the stimulus mentioned in this section? This should be clearly stated in the Methods section.

All instrument-based analyses must have sufficient detail on setting, etc. so that the analysis can be replicated in another laboratory.

Results

From the images provided in the figures, it appears that nearly all of the cells migrating through the membrane are mononuclear. Given the large size of mature osteoclasts, it is surprising that they migrated through a membrane. Pictures of the cells showing multinucleate cells with nuclear staining are needed. As further support for the identity of the active cells, in Figure 3a, it is clear that the highest intensity of red is associated with mononuclear cells. In Figure 3c most of the pits appear to be very small. What is the length of the size bar in the images in this figure? In addition,

it is well-known in the field that multinucleated osteoclasts are rarely released by the described treatment. What occurs is that TRAP-positive mononuclear cells, most likely capable of fusing to form osteoclasts, are lifted off. Figures 4 d and e could provide some approximation of the number of probable mononuclear cells and multinucleated cells in vivo. 4d would need to included a TRAP stain for this to be meaningful. It is possible that the supplemental data addresses this concern. This reviewer was unable to locate the supplemental figures for this manuscript. This has greatly impeded evaluation of the soundness of the data.

Figure 4g-j: Normal serum analysis for bone resorption are TRAP and CTX.

Discussion

Page 23 line 1: What is HO48?

Reviewer #2:

Remarks to the Author:

Summary: This study investigates a unique and interesting approach to removing heterotopic ossification that has already formed. The authors have developed a chemical approach to modify osteoclasts (bone resorbing cells) with tetracycline that enhances the ability of osteoclasts to bind to heterotopic bone surfaces and remove the ectopic bone tissue.

1. The first sentence of the Abstract (page 1, line 19) indicates that the only role of osteoclasts is the removal of ectopic calcification; this is not a true statement, and it does not appear to be the authors' intended message.

2. Page 1, Line 21. Heterotopic ossification is incorrectly described as "precipitation of calcium minerals". HO involves osteoblastic cell differentiation and bone tissue formation.

3. Page 2, Lines 15-16 to Page 3, Line 1. The authors state that osteoclasts poorly resorb ectopic bone. It is important that this statement is appropriately referenced since it supports the basic premise of the study.

4. The basic experimental rationale of the study is that tetracycline binds and coats the membrane surfaces of osteoclasts. While tetracycline binding to mineralized bone is established in the field, binding to osteoclast cell membranes is not established and is only supported in the current manuscript by citation of a 1997 reference that does not appear to have investigated tetracycline or osteoclasts.

The authors have provided a limited description of how or why tetracycline binds osteoclasts. The Introduction suggested that the approach is to add a mineral shell to cells, however tetracycline is also suggested to bind directly to components of the osteoclast cell membrane. A schematic (Fig 1b), but no data, is provided to illustrate the process of OC-TC binding.

5. Visual evidence of two single cells in vitro are provided as examples of the "tetracycline osteoclast cocoon" in Figure 1g. It is unclear what the authors intend to suggest by Figure 1b which implies that their TC-OC are present in bone in vivo; in vivo data were not shown. Figure 1b also suggests that there is an established and specific chemistry for TC binding to OC, however, as noted above, data were not shown. It is unclear whether TC will generally bind any cell membrane or is specific for osteoclasts, or if the osteoclasts have been modified. The stability of TC binding to osteoclasts in vivo is unclear.

6. Supplementary Figure 3: this in vitro assay does not provide evidence of osteoclast ability to bind or not bind soft tissues.

7. Figure 1h,i shows TEM of osteoclasts, but does not confirm that these cells are actively

resorbing bone.

8. Functional bone-resorbing osteoclasts are multi-nuclear, however the studies refer to non-TC bound osteoclasts as mononuclear – it is not clear that a relevant control cell population has been examined. (see Figure 2i)

9. The design of the experiment in Figure 3 is unclear. Cells are described as cultured with ectopic calcification, but the source of the calcification and how its quality and quantity were controlled in the assay was not described in Results or Methods. Additional information about how quantitative data in Figure 3 were acquired is needed (Were multiple fields of view examined per sample? How were regions selected?)

10. The Method of tendon injury to induce HO was not described.

11. For cell implant experiments in Figure 4, how was the viability of cells (pre-implant and in situ) evaluated? Was the localization and density of the injected cells verified/examined?

12. Quantitative assessments cannot be convincingly made based on the data in Figure 4d.

13. The images in Figure 4e cannot conclusively identify the implanted cells. Even so, the authors conclude that more OC-TC than OC are present after 2 hours. The authors are asked to explain where the implanted OC would be located if they are not present at the injection site.

14. In Figure 4 experiments, was skeletal bone in the area of cell implants evaluated for associated implanted osteoclasts and bone resorption?

15. As noted by the authors in the Discussion, bone resorption by osteoclasts is normally coupled to bone formation by osteoblasts. It would be of great interest to conduct bone formation assays in this system.

Additional comments:

The manuscript could be substantially improved by careful choice of wording and descriptions. In addition to points noted above, examples in Abstract and Introduction follow but this comment applies throughout the manuscript.

Page 1, Line 18. The description of osteoclast function is oversimplified by only mentioning decalcification activity.

Page 2, Line 7. "Bionic" does not seem to be the best word choice here since it implies a mechanical or electronic replacement.

Page 3, Line 12. Consideration should be given to not describing osteoclasts as "intelligent materials".

Page 3, Lines 10-11. The authors are requested to clarify under what circumstances yeast coated with mineral shells could be expected to increase survival.

Page 4, Line 5-6. Does tetracycline bind the osteoclasts or the bone mineral?

Reviewer #3:

Remarks to the Author:

So far, no effective methods have been developed to remove calcified tissue in Heterotopic Ossification (HO) sites. In this study of Jin W et al, osteoclasts (OCs) were induced from bone marrow cells and cell surface tetracycline (TC)-covalently modified OCs (TC-OCs) were engineered.

These TC-OCs were found to be viable and functional and have a stronger ability in targeting to bone surface compared to normal OCs. TC-OCs were found to have a stronger ability in resorbing bone in vitro and resorbing calcified tissues at HO site in a tendon tenotomy-induced HO model in rats. These findings suggest that artificially engineered osteoclasts can be developed as a potential novel living treatment for HO. The concept and findings are interesting. I have the following concerns.

1. There are so many different types of HOs, apart from the HO induced by tenotomy, efficacy of these engineered TC-OCs in resorbing calcified tissue should be demonstrated in another type of HO, which will increase the convincing nature of the findings.
2. Differences of TC-OCs should be investigated in resorbing calcified tissues present in endochondral bone (containing calcified cartilage) and present in intramembranous bone (containing no calcified cartilage).
3. Survival ability of TC-OCs vs OCs should be compared in vivo.
4. Conclusion "that the artificially engineered cells can be developed as the novel living treatment materials for biomedical therapy" is too broad and too general and cannot be supported by existing data (which is related to resorbing calcified tissue in HO site).
5. Methodology should be shortened.
6. Thorough proof reading/editing is required, and English language usage should be improved.
7. Errors in texts: Line 10 page 22: "First, OCs and immune cells are of bone marrow stromal cell origin"; Line 11 page 24: "Adherent cells were harvested as BMMs".

Cory J. Xian

Point-by-Point Response

Reviewer#1:

Comment-1: *“As written, the first sentence in the abstract is incorrect: “their unique decalcification activity is strictly limited to the removal of ectopic calcification”. It’s clear from the remainder of the abstract that the authors do not mean to state that osteoclasts can only remove ectopic calcification, which is what is stated. Please revise.”*

Response: Thanks for the advice. We have revised the first sentence in the abstract about the limitation of osteoclast function to treat HO to avoid misunderstanding.

The improved statement reads (Page 1, line 20), “Osteoclasts (OCs), the only cells capable of remodelling bone, can demineralize calcium minerals biologically. Naive OCs have limitations for the removal of ectopic calcification, such as in heterotopic ossification (HO), due to their restricted activity, migration and poor adhesion to sites of ectopic calcification.”

Comment-2: *“Osteoclast cultures. This is unnecessarily complex. There is no need to discuss the dose/response studies, since it is so well-documented that M-CSF and RANKL are required for osteoclast differentiation.”*

Response: The comment is reasonable. We have deleted the discussion of dose as the suggestion. Meanwhile, we have simplified the process of OCs differentiation.

Related results and methods now reads (Page 4, line 15): “In our study, OCs were induced by bone marrow-derived monocyte-macrophage precursor cells (BMMs) stimulated by receptor activator of nuclear factor- κ B ligand (RANKL, 50 μ g/l) and macrophage colony-stimulating factor (M-CSF, 25 μ g/l).”

Page 31, line 2-9: “**OC differentiation.** Six to eight-week-old male mice (C57BL/6, Shanghai SLAC Laboratory Animal Co., Ltd., China) were sacrificed by decapitation under deep anaesthesia with 10% chloral hydrate. Bone marrow cells(BMMs) were isolated from the tibiae and femurs and cultured in α -MEM (Gibco, USA) containing 25 ng/ml M-CSF (R&D system), 10% FBS (Gibco, USA), 100 U/ml penicillin G, and 100 lg/ml streptomycin for 24 h at 37 °C overnight under 5% CO₂. BMMs were seeded onto 24-well plates (4×10^4 cells/well) or 96-well plates (1×10^4 cells/well) and cultured for 6 days in α -MEM supplemented with 10% (v/v) FBS, 25 ng/ml M-CSF and 50 ng/ml RANKL (R&D system) at 37 °C with 5% CO₂. The culture medium was replaced with fresh medium every other day.”

Comment-3: “*Please define CLSM the first time it is used in the methods section. Is there a reason that two different sources are used for this? Please explain.*”

Response: Thanks for your careful check! We apologize for omitting the definition of CLSM at its first appearance. We have defined CLSM as suggested in the comment. CLSM the first time it is used in the methods section and now related methods now reads(Page 32, line 4) “The cells were then rinsed with PBS and seeded in 3.5-cm wells with coverslips before being subjected to confocal laser scanning microscopy (CLSM).”

The reason for the two different sources of CLSM is the limitation of our facility platform so that two CLSMs were arranged for the use alternately. However, we had tried our best to complete the similar experiments by using the same CLSM. For example, all CLSM images in **Fig. 1g** were from FV1000 and the images of TC modified on OCs cell membrane at day 1, 2, 3 in **Supplementary Fig. 4** were detected by Nikon. We are sorry for our negligence of unclear description the sources of CLSM in the previous version, which has been corrected in the present version.

In the methods section now reads (Page 32, line 7-9): “Then, the cells were rinsed with PBS three times and subjected to CLSM (FV1000, Olympus) at 0, 4 days. Subsequently, native OCs and TC-OCs were cultured with α -MEM (10% FBS) and then continuously subjected to CLSM (Nikon) at 1, 2, 3 days.”

Comment-4: *“Quantitating TC molecules on the osteoclasts: This is confusing. Are you making the assumption that all of the TC molecules in the lower concentrations of your standard curve attach to the cells? If so, please make this assumption clear or explain your quantitation more clearly.”*

Response: Thanks for the comments. We are sorry for not clearly describing the *quantitation* of TC molecules. The concentration of TC on the cell membrane was calculated by standard curves. Through the OD data for the TC-OC cell suspension solution, the concentration of TC-OCs was estimated.

To address this comment, we added the details of the quantitation in the Materials and Methods section (Page 33, line 2-14 to Page 34, line 1-3): “The number of TC molecules attached to the TC-OC cell membrane was quantified at working concentrations (160 μ g/ml) by Varioskan Flash (Thermo) based on their autofluorescence ability. TC-OCs were prepared (100 μ l, 2×10^5 cells/ml) and

resuspended in a 0.9% NaCl solution before being added to 96-well plates. The concentration of TC (the value is 20 µg/ml) on cell surface was obtained according to OD data (the value is 0.2) of solution measured by Microplate reader. The number of TC molecules on the cell membrane was calculated by using a standard linear calibration curve of TC solution (100 µl; 10 µg/ml, 20 µg/ml, 40 µg/ml, 80 µg/ml and 160 µg/ml).

The results showed that each cell was modified with approximately 1.35×10^{11} TC molecules.

$$C_{TC} = 20 \text{ µg/ml}; V = 100 \text{ µl}$$

$$m_{TC} = C \times V = 20 \text{ µg/ml} \times 0.1 \text{ ml} = 2 \text{ µg}$$

$$N_{total} = n \times N_A = \frac{N_A \times m_{TC}}{M_{TC}} = 6.02 \times 10^{23} \times 2 \times 10^{-6} / 444.5 \approx 2.7 \times 10^{15} \text{ molecules}$$

$$N_{tc/cell} = \frac{N_{total}}{N_{cell}} = 2.7 \times 10^{15} / 2 \times 10^4 \approx 1.35 \times 10^{11} \text{ molecules}$$

C_{TC} is the concentration of TC on the OC cell membrane surface; V is the volume of cell suspension; N_{total} is the total number of tetracycline molecules; N_A is the Avogadro's number; N_{cell} is number of OCs and $N_{tc/cell}$ is number of tetracycline molecules on every OC.”

Comment-5: *“Flow cytometry: osteoclasts are large cells and most are sheared if the aperture of the machine is not sufficiently large. Please provide more precise details of the flow protocol. Were the cells evaluated visually to determine how intact there were after flow?”*

Response: We agree with the comment that OCs are large cells. Cell engineering was indirectly estimated by flow cytometry. In our experiment, the surface engineered cells were filtered through 200

mesh cell filters (diameter is approximately 70 μm) before flow cytometry detection. Flow chamber size of flow cytometry (CytoFLEX LX, Beckman Coulter, US) is 430 \times 180 μm , which is sufficient to measure the fluorescence of TC. Besides, CLSM images and IR analysis (**Fig. 1f, g**) also proved the success of cell engineering.

To eliminate the referees' concerns about the precise details of the flow protocol, we have carefully improved the methods and added more details about instruments in the new manuscript. After repeated verification of flow cytometry analysis, OCs and engineered OCs were still intact.

The Flow cytometry section now reads (Page 34, line 5-16 to Page 35, line 1-3), "**Flow cytometry analysis.** To evaluate encapsulation efficiency, surface-engineered OC was stained with 250 nM PKH26, which is not membrane penetrable. Cell surface engineering efficiency was indicated by TC (160 $\mu\text{g}/\text{ml}$) autofluorescent on cell membranes. Mature OCs were digested by trypsin with 0.25% EDTA for about 20 mins and washed with PBS for three times. Then, OCs were encapsulated at different concentrations (0 $\mu\text{g}/\text{ml}$, 10 $\mu\text{g}/\text{ml}$, 20 $\mu\text{g}/\text{ml}$, 40 $\mu\text{g}/\text{ml}$, 80 $\mu\text{g}/\text{ml}$, 160 $\mu\text{g}/\text{ml}$ and 320 $\mu\text{g}/\text{ml}$) according to the surface engineering method described above. Control cells were stained with 250 nM PKH26 for general cell membrane labelling for 3 min and with 5 μM Hoechst 33258 for another 3 min, with no TC treatment. The stained cells were washed, collected, and adjusted to 300 μl of cell suspension solution (1×10^6 cell/ml), and then analysed on an CytoFLEX LX (Beckman Coulter, US, flow cell dimensions: 430 μm \times 180 μm internal diameter). The fluorescence of naive cells was used as a control and cells were filtered through 200 mesh cell filters. The linear gains were set in the light scatter channel, and the fluorescence channel was set on a logarithmic scale. 10,000 cells were analyzed in each condition. Every samples were conducted to three replicates and protected from light. Finally, the fluorescence intensity was measured by FlowJo software (FlowJo vX.0.7)."

Comment-6: *“Ectopic resorption: What is the stimulus mentioned in this section? This should be clearly stated in the Methods section.”*

Response: Thanks for the careful check! The stimulus is still M-CSF and RANKL. We have shown the detail of stimulus in the revised manuscripts. The relevant Methods now reads (Page 35, line 7): “OCs/TC-OCs (1×10^4 cells/well) were seeded onto bovine cortical bone slices and then cultured in α -MEM (10% FBS) with a stimulus (25 $\mu\text{g/l}$ M-CSF and 50 $\mu\text{g/l}$ RANKL) as described above for 6 days.”

Comment-7: *“AFM What is the stimulus mentioned in this section? This should be clearly stated in the Methods section.”*

Response: Thanks for the comment. We have provided more sufficient detail on stimulus mentioned in AFM experiments.

In the revised text, it now reads, page 10, line 6, “OCs/TC-OCs were separately seeded onto glass plates and cultured with α -MEM (10% FBS) supplemented with a stimulus (25 $\mu\text{g/l}$ M-CSF and 50 $\mu\text{g/l}$ RANKL) for 24 h.”

Page 38, line 13 “Then, phenol red-free α -MEM (Gibco, USA, 10% FBS) supplemented with 25 $\mu\text{g/l}$ M-CSF and 50 $\mu\text{g/l}$ RANKL was injected into the liquid cell.”

Comment-8: *“All instrument-bases analyses must have sufficient detail on setting, etc. so that the analysis can be replicated in another laboratory.”*

Response: We appreciate your recommendation to provide a more detailed description of the setting in the revised manuscript, including flow cytometry (Page 34, line 5), calcified tissue resorption (Page 35 line13), AFM experiment (Page 38 line 13) and HO animal model (Page 18 line 1).

Comment-9: *“From the images provided in the figures, it appears that nearly all of the cells migrating through the membrane are mononuclear. Given the large size of mature osteoclasts, it is surprising that they migrated through a membrane. Pictures of the cells showing multinucleate cells with nuclear staining are needed. As further support for the identity of the active cells, in Figure 3a, it is clear that the highest intensity of red is associated with mononuclear cells. In Figure 3c most of the pits appear to be very small.”*

Response: Thanks for the comment. Previous works had reported that osteoclasts could migrate through a membrane (Li, X., et al, *Osteoporosis Int.* 28.7 (2017): 2221-2231.). To further prove that the cells are multinucleated, cells that migrate to calcified tissue were stained with TRAP as the gold standard for OC identification. The results showed that the cells that migrated to calcified tissue were TRAP-positive cells (please see **Supplementary Fig. 9a, b**). More TRAP-positive cells in TC-OC groups migrated to calcified tissue compared with that in OC groups, suggesting that the migration ability of cells has been improved via cell surface engineering. As shown in **Figure 3a**, the structure associated with the strongest red signal is an irregular and incomplete cell structure, this red signal has no connection with mononuclear cells. The red signal intensity of multinucleated cells is the

quantitative basis.

SEM images show the surface information of erosive phagocytosis of calcified tissues but cannot reveal the depth of pit quantitatively. In order to address this problem, the depth of pits was quantitatively measured by TEM of bone slice and AFM additionally (**Supplementary Fig. 5d** and **Supplementary Fig. 6 b-d**). AFM images shows morphology of the pits and TEM shows the depth of the pits, the value was approximately 20 μm . Those results also confirmed that TC engineered OC exhibited excellent bone absorption capacity. It should be noted that this value is consistent with reports in the literature (Luo, J., *et al. Nat. Med.* **22**, 539 (2016)).

To address your comment, we added more discussion in new manuscripts (Page 9, line 5): “The OCs/TC-OCs resorption site on the cortical bone surface was quantitatively and qualitatively confirmed by transmission electron microscopy (TEM) and atomic force microscopy (AFM) (**Fig. 1h, i and Supplementary Fig. 5d and Supplementary Fig. 6**). The depth of pit was approximately 20 μm , which confirmed that cells had excellent bone resorption capacity. These results indicated that the TC-OCs and OCs shared the same calcium mineral resorption capacity, confirming that the fundamental function of OCs remained after TC modification.”

Comment-10: “*What is the length of the size bar in the images in this figure?*”

Response: Sorry for the missing the length of size bar. We have added the scale bar in **Figure 3** (please see page 17, line 1).

“**Figure 3** Increased ectopic calcified tissue resorption capacity of TC-OCs *in vitro*. (a) CLSM images of OCs/TC-OCs seeded into 48-well plates containing ectopic calcifications and incubated

with an AIETM pH probe (10^{-5} M) for 2 h. The images from panels A and B were merged. The arrow points to osteoclasts. Scale bar: 50 μ m. (b) Quantitative intensities of OCs/TC-OCs labelled with the AIETM pH probe ($n = 3$ independent samples/group). (c) SEM images of OCs and TC-OCs migrating into calcified tissue in vitro. Scale bar: 30 μ m. (d) SEM images of calcified tissue resorption area (three images taken from each sample and each group with four biological replicate samples). Scale bar: 300 μ m. (e) Statistical analysis shows the changes in the resorption area between the two groups. (f-h) Three-dimensional reconstruction of micro-CT scans of ectopic calcified tissue treated with 0.9% NaCl (blank), OCs, or TC-OCs at 0 and 2 weeks ($n = 3$ independent samples/group). Scale bar: 2 mm. (i) Resorption bone volume (BV) analysis showing the changes in resorption areas among the three groups. (j) Ca^{2+} eluted from the acid-etched areas in three groups, as measured by ICP-OES. (* $P < 0.0332$, ** $P < 0.0021$, *** $P < 0.0002$, **** $P < 0.0001$)”

Comment-11: *“In addition, it is well-known in the field that multinucleated osteoclasts are rarely released by the described treatment. What occurs is that TRAP-positive mononuclear cells, most likely capable of fusing to form osteoclasts, are lifted off. Figures 4 d and e could provide some approximation of the number of probable mononuclear cells and multinucleated cells in vivo.”*

Response: Thanks for the advices. You supposed that TRAP-positive mononuclear cells are capable of fusing to form OCs. To follow your advice, we quantified the number of mononuclear cells and multinucleated cells through TRAP staining images of calcified tissue in vitro (see **Supplementary Fig. 9a-d**). We postulate that the engineered cells were implanted in situ at ectopic calcification sites, and the number of probable mononuclear cells and multinucleated cells in vivo is consistent with that

in vitro. The number of mononuclear cells was 46 ± 10 in the OC group and 118 ± 17 in the TC-OC group. The number of multinucleated cells was 28 ± 3 in the OC group and 69 ± 6 in the TC-OC group. The number of monocytes was approximately 2.5-fold higher than the number of multinucleated cells in both the OC group and the TC-OC group. In general, this result revealed that calcified tissue-targeting ability of native OCs is significantly improved after modification with TC, allowing more multinucleated cells to adhere to calcified tissues.

Comment-12: *“4d would need to include a TRAP stain for this to be meaningful. It is possible that the supplemental data addresses this concern. This reviewer was unable to locate the supplemental figures for this manuscript. This has greatly impeded evaluation of the soundness of the data.”*

Response: Thanks for the advices. We have added an experiment f involving or TRAP staining of Tendon section. To follow your advices, we add more discussion about TRAP staining in the main text (please see page 19, line 9): “TRAP staining images and quantitative statistics demonstrated that more TRAP-positive cells were maintained and fewer ectopic calcifications were retained in TC-OC-treated rats than in the other rats. These pathological results demonstrated the effective and precise bone resorption of TC-OCs in the setting of HO.”

Comment-13: *“Figure 4g-j: Normal serum analysis for bone resorption are TRAP and CTX.”*

Response: The comment is constructive. We have added a new experiment involving serum analysis of the bone resorption-related factors TRAP and CTX (please see **Fig. 5l, m**). The TRAP and CTX is

6.31 ± 0.80 ng/ml and 0.21 ± 0.01 ng/ml in OC groups, while the values of engineered group are 7.42 ± 1.31 ng/ml and 0.21 ± 0.07 ng/ml. We found that the amount of TRAP and CTX for bone resorption in normal serum was unaffected by cell surface engineering. Thus, it can be ensured that the cytokine response will not appear in serum by using the engineered OCs.

To clarify this, the main text now reads (please see page 25, line 10-13): “Bone resorption by osteoclasts is normally coupled to bone formation by osteoblasts⁴⁶. tartrate-Resistant acid phosphatase (TRAP), type I collagen cross-linked C-telopeptides (CTX) were used as gold standard biomarkers for bone resorption. The serum levels of TRAP and CTX were normal (**Fig. 5m**), suggesting that surface engineering did not lead to side effect. Furthermore, no obvious changes were observed in blood chemistry or bone density (**Supplementary Fig. 14** and **Supplementary Fig. 15**) or in trabecular morphology, confirming the biocompatibility nature of TC-OCs in the treatment of HO.”

Comment-14: *“Page 23 line 1: What is HO48?”*

Response: Thanks for the careful check. We are really sorry about for typing mistake. We change the mistake in the main text to be correctly referenced. In the text, page 29, line 9: “Despite the vigorous regulation and control of bone equilibrium, changes in remodelling can occur as a result of imbalances between osteoclast and osteoblast activity, which results in debilitating bone diseases such as osteoporosis and HO.”

Reviewer#2:

Comment-1: *“The first sentence of the Abstract (page 1, line 19) indicates that the only role of osteoclasts is the removal of ectopic calcification; this is not a true statement, and it does not appear to be the authors’ intended message.”*

Response: We appreciate your recommendation. We have revised the statement about osteoclast biology and function. To eliminate misunderstanding, we add a new statement to the abstract (page 1, line 20): “Osteoclasts (OCs), the only cells capable of remodelling bone, can demineralize calcium minerals biologically.”

Comment-2: *“Page 1, Line 21. Heterotopic ossification is incorrectly described as “precipitation of calcium minerals”. HO involves osteoblastic cell differentiation and bone tissue formation.”*

Response: We agree with this comment. We have revised the statement about HO characteristics. The improved sentence now reads (page 2, line 3), “HO is the formation of pathological mature bone within extraskelatal soft tissues, and there are currently no reliable methods for removing these unexpected calcified plaques.”

Comment-3. *“Page 2, Lines 15-16 to Page 3, Line 1. The authors state that osteoclasts poorly resorb ectopic bone. It is important that this statement is appropriately referenced since it supports the basic premise of the study.”*

Response: Thanks for your valuable advice. We have added the related references to support the basic premise of its poor activity and resorption in ectopic bone (Doherty, T.M., *et al. FASEB. J.* **16**, 577 (2002); Massy, Z.A., *et al. Diabetes. Metab.*, **34**, 0-20 (2008)). In our study, adhesive and targeting capacity of cells was also increased by surface engineering, thereby improving their poor bone resorption capacity. Meanwhile, the poor targeting ability of osteoclasts to calcified tissues has also been further verified in our experiments (please see **Fig. 2** and **Supplementary Fig. 2**).

Comment-4. *“The basic experimental rationale of the study is that tetracycline binds and coats the membrane surfaces of osteoclasts. While tetracycline binding to mineralized bone is established in the field, binding to osteoclast cell membranes is not established and is only supported in the current manuscript by citation of a 1997 reference that does not appear to have investigated tetracycline or osteoclasts. The authors have provided a limited description of how or why tetracycline binds osteoclasts. The Introduction suggested that the approach is to add a mineral shell to cells, however tetracycline is also suggested to bind directly to components of the osteoclast cell membrane. A schematic (Fig 1b), but no data, is provided to illustrate the process of OC-TC binding.”*

Response: Thank for your valuable and thoughtful comments. First, the concern is the inadequate description and literature on TC binding to OC cell membranes. Previous work reported that the glycan chains on the surface of the cell membrane contained sialic acid and uronic acid, which exposed some carboxyl groups and laid a foundation for subsequent cell engineering (Takahata, M., *et al. Bone* **41**, 77-86 (2007); Matsumoto, *et al. A. J. Am. Chem. Soc.* **131**, 12022-12023 (2009)). The latest study also

suggested that liposomes with a similar structure to the cell membrane could be surface engineered by TC to target bone (Lin, X., *et al. J. Am. Chem. Soc.* **142**, 17543-17556 (2020)).

Second, there are no data to illustrate the process of OC-TC binding. We are sorry for not clearly describing the evidence of OC modification. In the present version, we have improved the statement to make this issue more clearly. In fact, the process of OC-TC binding to cell surface membrane was supported by IR analysis in previous **Supplementary Fig. 6, (Fig. 1f in revised manuscript)**. In order to enhance our evidence, we added Raman analysis to further prove the process of TC-OC binding. The results indicated that OCs were successfully engineered by the addition of TC (**Supplementary Fig. 4a**).

Page 5, line 16, we insert new discussion about the process of surface engineering. “Furthermore, the surface modification process was also detected by Roman spectroscopy, and the performance of the TC peak (1639 cm^{-1}) indicated the successful modification of the cell surface (**Supplementary Fig. 4a**).”

Comment-5. *“Visual evidence of two single cells in vitro are provided as examples of the “tetracycline osteoclast cocoon” in Figure 1g. It is unclear what the authors intend to suggest by Figure 1b which implies that their TC-OC are present in bone in vivo; in vivo data were not shown. Figure 1b also suggests that there is an established and specific chemistry for TC binding to OC, however, as noted above, data were not shown. It is unclear whether TC will generally bind any cell membrane or is specific for osteoclasts, or if the osteoclasts have been modified. The stability of TC binding to osteoclasts in vivo is unclear.”*

Response: I appreciate the reviewer's efforts to point out unclear statements and the need for additional experiments. In our study, we proposed a rational modification of OCs utilizing the chemical tetracycline (TC) to confer it the ability to adhere to and target ectopic calcification sites. Figure 1b represents for the scheme of our concept that the OC cell membrane was modified with TC molecules via conjugation of the amino group on TC to the hydroxyl group on the cell membrane surface. In addition, the adhesion of TC-modified cells to calcified tissues has improved due to the ability of TC to chelate calcium ions in calcified tissues. IR and Raman analyses of the process of OC-TC binding to the cell surface membrane are provided in the revised manuscript (**Fig. 1f**) and supplementary file (**Supplementary Fig. 4a**). To address your concerns, we added CLSM images to prove stability of TC binding to osteoclasts in vivo (**Supplementary Fig. 11**). The CLSM images directly proved that TC stably present on osteoclasts cell surface membrane for more than 4 days in vivo.

In order to eliminate the concern, we add new discussion in main text (**Page 18, line 2**): "The timeline of injection depends on stability of TC on cell surface in vivo. The engineered OCs with TC molecules on the cell surface can be stored stably in vivo over 4 days (**Supplementary Fig. 11**)."

Comment-6. *"Supplementary Figure 3: this in vitro assay does not provide evidence of osteoclast ability to bind or not bind soft tissues."*

Response: Thank you for the valuable and thoughtful comment. Evidence of the ability of OCs to bind soft tissues in vitro was added to the new supporting information (please see **Supplementary Fig. 2**). The results show that the ability of osteoclast bind to soft tissues is weak.

To address your concern, we add the short sentence in the new manuscript (**Page 5, line 6**): “As expected, their adhesion ability within ectopic calcified tissue and soft tissues was poor (**Supplementary Fig. 2**), which was also confirmed by previous study.”

Comment-7. *“Figure 1h, i shows osteoclasts, but does not confirm that these cells are actively resorbing bone.”*

Response: This concern is reasonable. After improving the experimental method with more attempts, we have observed that both TC-OCs and native OCs were actively absorbing bone (**Figure 1h, i**). The new images also indicate that the surface-engineered OCs showed no changes in their vital function of bone erosion.

Comment-8. *“Functional bone-resorbing osteoclasts are multi-nuclear, however the studies refer to non-TC bound osteoclasts as mononuclear – it is not clear that a relevant control cell population has been examined. (see Figure 2i)”*

Response: We apologize for our negligence in describing OCs as mononuclear. We have examined cell population via CLSM. As depicted in **Fig. 2j, k**, CLSM images results further confirmed that the cells attached to the tipless cantilever were multinucleate. We have revised mistakes in the results section. Now it reads (**Page 11, line 9**), “Moreover, CLSM results further confirmed that the cells attached to the tipless cantilever were multinucleate (**Fig. 2j, k**).”

Comment-9. *“The design of the experiment in Figure 3 is unclear. Cells are described as cultured*

with ectopic calcification, but the source of the calcification and how its quality and quantity were controlled in the assay was not described in Results or Methods. Additional information about how quantitative data in Figure 3 were acquired is needed (Were multiple fields of view examined per sample? How were regions selected?)”

Response: Thank you for your constructive suggestions. We apologize for not describing the source of the calcification and the qualitative and quantitative methods clearly. The detailed source of the calcification and quantitative data in Figure 3 had been added in the improved manuscripts.

Calcified tissues were derived from an animal model of tenotomy-induced heterotopic ossification. The BV of the ectopic calcified tissue was measured by micro-CT, and the 9 calcified tissues were randomly divided into three groups, with three biological replicate samples in each group. The initial BV values of each group were $28.08 \pm 6.01 \text{ mm}^3$ (blank), $31.42 \pm 9.21 \text{ mm}^3$ (OCs), and $28.55 \pm 3.65 \text{ mm}^3$ (TC-OCs), respectively. The bone resorption rate is obtained by the bone volume of 0 and 2 weeks. The formula is:

$$\text{Bone resorption ratio} = \frac{BV(0 \text{ week}) - BV(2 \text{ weeks})}{BV(0 \text{ week})} \times 100\%$$

Ca^{2+} eluted from the acid-etched areas as measured by ICP-OES. And the concentration of Ca^{2+} at 0 week is $48.47 \pm 1.08 \text{ mg/l}$ in blank, $45.81 \pm 2.99 \text{ mg/l}$ in OC groups and $43.03 \pm 2.37 \text{ mg/l}$ in TC-OC groups. The concentration of Ca^{2+} eluted from the acid-etched areas of culture media was obtained by this formula: $C_{\text{Ca}}(\text{elution}) = C_{\text{Ca}}(2 \text{ week}) - C_{\text{Ca}}(0 \text{ week})$. Then, the total of Ca eluted from the acid-etched areas was obtained, the value is $63.05 \pm 6.03 \text{ mg/l}$ in blank, $81.67 \pm 13.22 \text{ mg/l}$ in OC groups and $170.57 \pm 30.78 \text{ mg/l}$.

Thank you for your comments. Three images taken from each sample and each group with four biological replicate samples. The regions were selected randomly and representative images of per groups are shown. We have explained this in detail in the new version of the manuscript according to your comments. Please see page17, line 6: “SEM images of calcified tissue resorption area (three images taken from each sample and each group with four biological replicate samples). Scale bar: 300 μm . ” The fluorescence intensity of the pH probes in three groups and the acid phagocytosis area of the calcified tissues were calculated by Image-J.

According to your helpful advice. We have improved the discussion and methods in Figure 3 and now improved methods reads (page 35, line 13 to page 37, line 7), “To assess calcified tissue resorption *in vitro*, eight-week-old male Sprague-Dawley rats (body weight 290-330 g) were preformed tenotomy-induced HO. Rats were maintained at Sir Run Run Shaw Hospital, Zhejiang University School of Medicine. Specific animal model methods were listed in the section of Animal Model section in the supplementary file. After routine feeding and observation for 12 weeks, the animals were subjected to X-ray analysis (Faxitron MX-20, USA) and micro-CT (Suzhou Hiscan Information Technology Co., Ltd.). Quantitative analysis of calcified tissue BV by micro-CT. Nine calcified tissue samples were randomly divided into three groups: blank (0.9% NaCl), OCs, and TC-OCs. The BV values were $28.08 \pm 6.01 \text{ mm}^3$, $31.42 \pm 9.21 \text{ mm}^3$, and $28.55 \pm 3.65 \text{ mm}^3$ before cell treatment, respectively. Then, OCs or TC-OCs (2×10^4 per well) were seeded onto one side of a well in a 48-well plate, and ectopic calcified tissue was seeded on the other side of the well. Then, cells were were seeded every two days and cultured in α -MEM (10% FBS) with 25 $\mu\text{g/l}$ M-CSF and 50 $\mu\text{g/l}$ RANKL. The BV of calcified tissues after TC-OCs treatment was obtained by Micro-CT in 2 weeks. The BV resorption rate of calcified tissue is calculated from the BV of calcified tissue before OCs-treatment (0 weeks) and the

BV after treatment (2 weeks). The formula is

$$\frac{BV(0 \text{ week}) - BV(2 \text{ weeks})}{BV(0 \text{ week})} \times 100\%$$

For quantitative analysis of the bone resorption capacity of OCs/TC-OCs *in vitro*, Ca²⁺ eluted from the acid-etched areas of culture media in 48-well plates (1 ml of 10% FBS α -MEM) was analysed by inductively coupled plasma optical emission spectrometry (ICP-OES) (Thermo iCAP 10 6300, USA).

The medium was diluted to 10 ml, and an analytical calcium standard solution (Aladdin Company) of 100.00 \pm 0.10 μ g/ml in 5% nitric acid was used as a stock solution for calibration. All solutions were filtered through 0.22- μ m Millipore films prior to use. And the concentration of Ca²⁺ at 0 week is 48.47 \pm 1.08 mg/l in blank, 45.81 \pm 2.99 mg/l in OC groups and 43.03 \pm 2.37 mg/l in TC-OC groups. Then, the total of Ca was obtained at 2 weeks, the value is 63.05 \pm 6.03 mg/l in blank, 81.67 \pm 13.22 mg/l in OC groups and 170.57 \pm 30.78 mg/l. The concentration of Ca²⁺ eluted from the acid-etched areas of culture media was obtained through those data. The formula is: C_{Ca}(elution) = C_{Ca}(2 week) - C_{Ca}(0 week). The amount of Ca²⁺ eluted from the acid-etched areas into the culture medium was determined by ICP-OES, and the value were 35.85 \pm 16.10 mg/l, and 127.54 \pm 33.07 mg/l.”

The related discussion is also improved accordingly and now it reads, “Microcomputed tomography (micro-CT) reconstruction was performed on ectopic calcified tissue samples to assess decalcification at the histological level (**Fig. 3f-h**). The volume of calcification at 0 and 2 weeks was quantitatively estimated by micro-CT. The percentage of bone volume reduction was used to evaluate the efficiency of bone resorption by engineered osteoclasts (please see methods). The total ectopic calcification volumes in the OC and TC-OC groups were reduced by 33.7 \pm 4.6 % and 67.0 \pm 1.6%, respectively (**Fig. 3i**), compared with that observed in the control group (0.7 \pm 3.7 %). Because of the

increased cell migration, the amount of Ca^{2+} eluted from the acid-etched areas estimated by ICP-OES was used to compare the bone resorption capability between the native OCs and the modified cells. The amount of Ca^{2+} eluted from the acid-etched areas into the culture medium in modified OC groups was significantly increased by about three times (**Fig. 3j**) compared with native OC groups, verifying that a significantly improvement in ectopic calcified tissue resorption capacity after surface modification. All these in vitro results provide direct evidence of the advantages of TC-OCs in biological decalcification”

Comment-10. *“The Method of tendon injury to induce HO was not described.”*

Response: Thanks for the comment, which noted that we have omitted the important information about the method of HO animal model. Actually, we described the animal modeling method in the “Animal model” section of the supplementary material in the previous version. In the present version, we have improved a statement to clarify this issue. Please see page 18, line 1.

In the revised text, page 18, line 1 it now reads: “The detailed methods of standard tenotomy rat model were provided in the supplementary files.”

Comment-11. *“For cell implant experiments in Figure 4, how was the viability of cells (pre-implant and in situ) evaluated? Was the localization and density of the injected cells verified/examined?”*

Response: Thanks for the thoughtful comments. The first concern is the viability of cells pre-implanted and in situ cells. Actually, the modification process had good compatibility with OCs, which

is proved by CCK8 assay (**Fig. 1c**) and CLSM images in **Fig 1g**. To dissipate the concern and follow the constructive suggestions to improve our work further, we have added a new experiment to examine of localization and density of the injected cells. The location and shape of the ectopic calcifications and the density of OCs/TC-OCs in each Achilles tendon were clearly detected under two-photon CLSM at 0 and 2 h after injection. It is concluded that the engineered OCs have a superior adhesion and calcification-targeting ability superior to that of naive OCs in vivo.

We also add a sentence in the main text (page 21, line 6-16) to make the statements more clearly to address this concern, which reads, “To further examine the ability of TC-OCs to target and adhere to ectopic calcification in vivo compared with OCs, cells were injected in situ injected into the Achilles tendons of the model rats⁴⁴, and these tendons were dissected after 0 h and 2 h. For the specific method of tendon injury used to induce HO, please refer to the supplementary file. As shown in **Fig. 5a-c**, at high spatial distribution, deep tissues were visualized at various depths with a view up to 100 μm deep by two-photon CLSM. The location and shape of the ectopic calcifications and the density of OCs/TC-OCs in each Achilles tendon were clearly detected under two-photon CLSM at 0 h in the three groups (**Fig. 5a-c**). CLSM images of the OCs/TC-OCs distribution in the calcification in tendon were captured at 2 hours after injection (**Fig. d-f**). The number of TC-OCs at the calcification sites was increased compared with that in the OC group, with values of 152 ± 30 in the TC-OC groups, 82 ± 17 in OC groups (**Fig. 5g**), implying that the engineered OCs have an adhesion and calcification-targeting ability superior to that of naive OCs.”

Comment-12. *“Quantitative assessments cannot be convincingly made based on the data in Figure 4d.”*

Response: Thank you for the instructive suggestion. The viability and density of pre-implant cells was qualitatively and quantitatively verified by two-photon CLSM (please see **Fig.5a**) to ensure cell viability was parallel at the initial. At the same time, we detected the location and density of the cells at the calcification site after injection 2 h, and multiple fields of view were observed (n=4) (see **Fig.5g**). After the improvement of the experimental method, we believe the current data is credible.

In order to eliminate your concern, we add a sentence in the main text to make the statement more clearly (page 21, line 6-16), which reads, “To further examine the ability of TC-OCs to target and adhere to ectopic calcification in vivo compared with OCs, cells were injected in situ injected into the Achilles tendons of the model rats⁴⁴, and these tendons were dissected after 0 h and 2 h. For the specific method of tendon injury used to induce HO, please refer to the supplementary file. As shown in **Fig. 5a-c**, at high spatial distribution, deep tissues were visualized at various depths with a view up to 100 μm deep by two-photon CLSM. The location and shape of the ectopic calcifications and the density of OCs/TC-OCs in each Achilles tendon were clearly detected under two-photon CLSM at 0 h in the three groups (**Fig. 5a-c**). CLSM images of the OCs/TC-OCs distribution in the calcification in tendon were captured at 2 hours after injection (**Fig. d-f**). The number of TC-OCs at the calcification sites was increased compared with that in the OC group, with values of 152 ± 30 in the TC-OC groups, 82 ± 17 in OC groups (**Fig. 5g**), implying that the engineered OCs have an adhesion and calcification-targeting ability superior to that of naive OCs.”

Comment-13. *“The images in Figure 4e cannot conclusively identify the implanted cells. Even so, the authors conclude that more OC-TC than OC are present after 2 hours. The authors are asked to explain*

where the implanted OC would be located if they are not present at the injection site.”

Response: Thank you for your valuable comments about the location of the implanted OCs. First, our experimental data proved that the adhesive ability of natural osteoclasts is weaker than that of engineered osteoclasts (see **Fig. 2**). Therefore, more OC-TC than OC are present after 2 hours. To address your comments, we have improved the experimental methods about the location of implanted cell. We visualized and quantified the location of the native and engineered OCs labeled by Hoechst 33258 in vivo at 0 and 2 h based on the CLSM images (**Fig. 5a**), and the results showed that the implanted the OCs presented in the tendon tissue calcification site not adhered to the calcified site weak ability to target calcified tissues (the cell number is 82 ± 17) after 2 h. In contrast, TC-OCs presented powerful adhesion and gather to the calcified site due to the chelating ability of cell surface engineered molecules to calcified tissue (the cell number is 152 ± 30) after 2 h. Next, the long-term (4 day) fate of the engineered OCs in vivo was conducted by flow cytometry.

Flow cytometry quantitative analysis further revealed that the engineered OCs present higher cell viability compared with natural OCs, the value is 27.56 ± 7.70 % in OC groups, 76.62 ± 9.99 % in TC-OCs groups, respectively (please see **Fig. 5k**). Natural OCs run off after long-term circulation owing to weaker cell adhesion and targeting. All these results demonstrated that the importance of surface engineering is essential for natural OCs to maintain the key functions of bone resorption in vivo for a long time. Long-term of circulation in vivo further demonstrated the favourable bone resorption activity of TC-engineered OCs and the feasibility of HO treatment.

Comment-14. *“In Figure 4 experiments, was skeletal bone in the area of cell implants evaluated for*

associated implanted osteoclasts and bone resorption? As noted by the authors in the Discussion, bone resorption by osteoclasts is normally coupled to bone formation by osteoblasts. It would be of great interest to conduct bone formation assays in this system.”

Response: This is a good question, thanks for the valuable advice. Based on an Elisa experiment, we determined the levels of TRAP and CTX related to bone resorption in Achilles tendon lysate (see **Fig. 5o**). The TRAP and CTX is 30.86 ± 4.60 ng/ml and 1.49 ± 0.17 ng/ml in blank group, 35.96 ± 7.07 ng/ml and 2.07 ± 0.12 ng/ml in native OCs group, while the values of engineered group are 50.59 ± 2.57 ng/ml and 2.37 ± 0.05 ng/ml. The tendon lysate enzyme levels of TRAP and CTX increased compared with those of the control groups, which confirmed the bone resorption activity was improved after modification.

We further addressed the expression of osteogenic factors in serum and tendon lysate during reversal treatment of HO with modified osteoclasts in Discussion section according to your thoughtful comments (please see **Fig. 5n**). The results of ALP and BMP-2 are 18.99 ± 2.28 ng/ml and 6.35 ± 0.43 ng/ml in blank group, 14.22 ± 2.76 ng/ml and 4.89 ± 0.72 ng/ml in native group, 12.43 ± 0.36 ng/ml and 4.54 ± 0.22 ng/ml in engineered group. The concentration of osteogenesis-related factors (ALP and BMP-2) has decreased compared with the native OCs groups. Those results indicated that the surface engineering osteoclasts have excellent activity in bone resorption, confirming the potential of this new approach in HO-reversing treatment.

In order to eliminate the concern, we add new discussion in main text (page 26, line 1-10): “The biocompatibility and activity of the engineered OCs in vivo was further proven by testing cytokine response in tendon lysis. The cytokine response of bone formation and resorption in the area of the

cell implants was measured by ELISA. ALP, BMP-2, TRAP, CTX were used as gold standard biomarkers for bone formation and resorption. Although the expression of proteins associated with bone formation (ALP and BMP-2) in the area of the cell implant is lower than that in the control groups (**Fig. 5n**), proteins related to bone resorption (CTX and TRAP) were more highly expressed in the TC-OC groups (**Fig. 5o**). These results indicated that surface engineering leads to slight reduction of bone formation marker and presents good biocompatibility and bone resorption activity. The expression of proteins associated with bone resorption had improved, thus confirming the conceptual potential of this new approach to HO-reversing treatment.”

Comment-15. *“Page 1, Line 18. The description of osteoclast function is oversimplified by only mentioning decalcification activity.”*

Response: Thank you for the constructive suggestion. According to the valuable comment, we have corrected the descriptions of osteoclast function. Furthermore, we have had the manuscript polished with professional assistance in statement. In the revised abstract (page 1, line 20), it reads, “Osteoclasts (OCs), the only cells capable of remodelling bone, can demineralize calcium minerals biologically.”

Comment-16. *“Page 2, Line 7. “Bionic” does not seem to be the best word choice here since it implies a mechanical or electronic replacement.”*

Response: Thanks for the suggestion. However, we feel that the mechanical or electronic replacement is incorrect. As a proof of concept, our strategy provides an alternative tactic for the treatment of

reversible ectopic calcifications by modifying living cells; therefore, we feel that “cell therapy” should be more suitable, which is used to replace “bionic therapy” in the comment.

In the main text, this sentence now reads (page 2, line 5): “This achievement indicates that HO can be reversed using modified OCs and holds promise for engineering cells as “living treatment agents” for cell therapy.”

Comment-17. *“Page 3, Line 12. Consideration should be given to not describing osteoclasts as “intelligent materials”.”*

Response: Thank you for the constructive suggestion about the inappropriate description of osteoclast biology. In the revised manuscript, we use “intelligent bone-resorbing cells” instead of “intelligent materials”. In the main text, this sentence now reads (page 2, line 13): “As naturally intelligent bone resorbing cells, OCs can remove large amounts of calcium, phosphate and collagen fragments released by the dissolution of bone minerals and organic bone matrix during bone resorption, which would be toxic to cells.”

Comment-18. *“Page 3, Lines 10-11. The authors are requested to clarify under what circumstances yeast coated with mineral shells could be expected to increase survival.”*

Response: Thanks for the constructive advices. The main text has been revised, now it reads (page 3, line 11): “In 2008, our laboratory first proposed that yeast, could be coated with calcium phosphate

shells via a layer-by-layer (LBL) strategy, which is an effective method for cell surface modification, to enhance survivability.¹⁶”

Comment-19. “Page 4, Line 5-6. Does tetracycline bind the osteoclasts or the bone mineral?”

Response: Yes, TC can bind the osteoclasts or the bone mineral. First, TC was confirmed to be a bone-targeting molecule by a large number of previous studies (Molly, S., et al. *Int. J. Mol. Sci.* **18**, 1345 (2017); Yuan, H., et al. *Int. J. Nanomed.*, 5671 (2015)). Second, the OC cell membrane was modified with TC molecules through conjugation of the amino group on TC with the hydroxyl group on the cell membrane surface, as shown in **Fig. 1f and Supplementary Fig. 4a**. Our concept is that site-specific delivery of engineered OCs represents a therapeutic approach for treatments reversing heterotopic ossification. The engineered OCs did not affect the BMD of normal bone, which was confirmed by **Supplementary Fig. 15**.

Reviewer#3:

Comment-1. *“There are so many different types of HOs, apart from the HO induced by tenotomy, efficacy of these engineered TC-OCs in resorbing calcified tissue should be demonstrated in another type of HO, which will increase the convincing nature of the findings.”*

Response: Thank you for the valuable suggestions. It is right that HO is divided into two categories, the acquired non-genetic form and inherited genetic form. (refer...) The two common types of non-genetic HO are musculoskeletal trauma-induced HO and neurogenic trauma-induced HO. Inherited genetic HO is divided into fibrodysplasia ossificans progressiva (FOP) undergoes endochondral ossification, while progressive osseous heteroplasia (POH) and Albright inherited osteodystrophy (AHO) are HO through intramembranous ossification. In a word, these different types of HO have a common feature that contains calcium deposits, which is in line with our concept of using engineered OCs to chelate calcium to increase cell adhesion to calcified tissues. It is worth mentioning that the musculoskeletal trauma-induced HO selected in our study is one of the most typical HO and also widely studied by other scholars (Seavey, J.G., et al. *J. Orthop. Res.* **35**, 2397-2406 (2017).; Qureshi, A., et al. *Am. J. Pathol.* 187 (2017).). Therefore, this surface engineered osteoclasts are expected to resorb other types of HO. In the newly submitted manuscripts, we have improved our descriptions about HO model according to your helpful suggestion. **Page 17, line 16:** “Given the enhanced ectopic calcified tissue resorption capacity of TC-OCs, a standard, extensive, representative and animal model of tenotomy induced HO (**Supplementary Fig. 10**) was used to observe the function of TC-OCs in vivo.”

However, more evidence is needed to verify this speculation. We have tried our best to follow the advice the efficacy of engineered TC-OCs in resorbing calcified tissue was performed in cranial bone and tibia bone in vitro (please see **Supplementary Fig. 17 and Supplementary Fig. 18**). Compared with osteoclasts, engineered osteoclasts exhibit superior bone resorption capacity in vitro, and are presented in different types of bone. This new result follows that our technology is potential for other type HO. Although the strategy may be valid for other calcified tissues such as vascular calcification, more studies should be required to finally confirm such an idea. However, in the present study, we first suggest a concept-of-proof of that the engineered-OC can be used as a novel therapeutic agent for reversible calcification and we believe that much more studies should be subsequently perform in details to expand its potential application. Anyway, we feel that the present manuscript only provides a start of such studies.

Comment-2. *“Differences of TC-OCs should be investigated in resorbing calcified tissues present in endochondral bone (containing calcified cartilage) and present in intramembranous bone (containing no calcified cartilage).”*

Response: We agree with this constructive comment. However, the animal model of calcified tissues presented in endochondral bone (containing calcified cartilage) and presented in intramembranous bone (containing no calcified cartilage) requires a long experimental period, and we would not be able to respond to your concern in time. Compared with the calcified tissues presented in endochondral bone and in intramembranous bone, endochondral and intramembranous bone have a common feature that contains calcium deposits, which is in line with our concept of using engineered OCs to chelate calcium to increase cell adhesion to calcified tissues. To address the comment, we have added new

experiments on the capacity of TC-OCs bone resorption to endochondral bone and intramembranous bone in vitro. The efficiency of bone resorption was measured by Micro-CT and SEM (please see **Supplementary Fig. 17 and Supplementary Fig. 18**). As depicted in **Supplementary Fig. 17**, The BV/TV value decreased from 0.015 ± 0.002 % to 0.011 ± 0.001 % in the OC groups but the BV/TV value decreased from $0.015 \pm 2.34e-4$ % to $0.008 \pm 1.66e-4$ % in TC-OC groups. The BMD value decreased from 4216.02 ± 91.19 units to 3920.79 ± 72.71 units in the OC groups, but the value decreased from 4442.20 ± 114.97 to 3811.81 ± 69.47 in the TC-OC groups.

As depicted in **Supplementary Fig. 18**, BV/TV value in the TC-OC decreased from 0.36 ± 0.03 % to 0.28 ± 0.01 %, but the BV/TV value in OCs group decreased from 0.35 ± 0.02 % to 0.32 ± 0.02 %. BS/TV was value in the TC-OC groups decreased from 1.46 ± 0.16 mm⁻¹ to 1.05 ± 0.11 mm⁻¹, but the BS/TV value in OCs group decreased from 1.42 ± 0.07 mm⁻¹ to 1.26 ± 0.05 mm⁻¹. Compared with OCs, engineered OCs exhibit superior bone resorption capacity in vitro, and are present in different types of bone, which shows that this technology has potential for other types of HO. These results indicated that TC-engineered OCs promote excellent ability of bone resorption regardless of HO type. Therefore, surface-modified osteoclasts have the potential not only for HO-reversing treatment but also for the treatment of other types of ectopic calcification.

Accordingly, we have added a new discussion in main text (page 27, line 5-16): “To further prove the bone resorption capacity of engineered TC-OCs in other calcified tissue, cells were applied to resorb in endochondral bone (cranial bone) and intramembranous bone (tibial bone) in vitro. The histomorphometric parameters bone volume (BV)/ total volume (TV), bone mean density (BMD, Hounsfield Units) and Bone surface (BS)/TV were examined by micro-CT. As picture of **Supplementary Fig. 17**, the BMD and BV/TV of cranial bone in TC-OCs group were lower than

those of the blank and OCs groups. BS/TV as the standard for analysis of the bone surface density were used to estimate TC-engineered OCs bone resorption capacity to absorb tibial bone. As depicted in **Supplementary Fig. 18**, BS/TV and BV/TV were decreased in TC-OC groups compared with OC-treated groups, confirming that TC-engineered OCs promote excellent ability of bone resorption in endochondral bone (cranial bone) and intramembranous bone. These data support the notion that surface-modified osteoclasts have the potential not only for HO-reversing treatment but also for the treatment of other types of ectopic calcification.”

Comment-3. “*Survival ability of TC-OCs vs OCs should be compared in vivo.*”

Response: Thanks for the advice. We have added two tests to qualitatively and quantitatively investigate the survival ability of TC-OCs and OCs *in vivo*.

1) Qualitative evaluations of OCs and TC-engineered OC cell viability *in vivo* were performed by Bio-Real *in vivo* imaging (please see **Supplementary Fig. 13**). We visualized the cell viability of native and engineered OCs labelled by Cell Trace Far Red DDAO-SE fluorescent tag in tendons based on the *in vivo* living imaging at 0 day and 4 days. The results showed that labelled TC-OCs showed a stronger staining intensity than OCs. Thus, it can be ensured that the survival ability of TC-OCs is better than OCs *in vivo* after 4 days, confirming the proof of this new concept.

2) Quantitative analysis of OCs and TC-engineered OCs cell viability *in vivo* was accomplished by flow cytometry. (please see **Fig. 5h-j**). Flow cytometry further revealed increased fluorescence signals of Calcein AM in the TC-OC groups (**Fig. 5k**), and the values were improved from 27.56 ± 7.70 % in OCs groups to 76.62 ± 9.99 % in TC-OCs groups. The results demonstrated that surface engineering

is essential for natural OCs to maintain the key functions of bone resorption in vivo for long-term circulation.

In the main text, we added a new statement accordingly and it reads (page 22, line 5-11 and page 24, line 9-15), “To qualitatively and quantitatively evaluate the cell viability of osteoclasts and TC-engineered osteoclasts in vivo, OCs with or without surface modification were used to perform in situ injections for an extended time period. We visualized the cell viability of native and engineered OCs labelled by Cell Trace Far Red DDAO-SE fluorescent tag in vivo based on the in vivo living imaging at 0 day and 4 days (**Supplementary Fig. 13**). The results showed that labelled TC-OCs showed a stronger staining intensity than OCs. The long-term in vivo circulation of the engineered OCs in ectopic tissue demonstrated that the TC-OCs retain good cell viability and function for more than 4 days.

Moreover, we quantified the cell viability of native and engineered OCs in vivo by flow cytometry at 0 day and 4 days after injection. Flow cytometry further revealed increased fluorescence signals of Calcein AM in the TC-OC groups (**Fig. 5h-j**), and the values were improved from 27.56 ± 7.70 % in OCs groups to 76.62 ± 9.99 % in TC-OCs groups. According to the improvement of cell viability with the modification of the cell membrane (**Fig. 5k**), we deduced that the TC-engineered OCs increase their unique function of bone resorption. The results demonstrated that surface engineering is essential for natural OCs to maintain the key functions of bone resorption in vivo for long-term circulation. Long-term of circulation in vivo further demonstrated the favourable bone resorption activity of TC-engineered OCs and the feasibility of HO treatment.”

Comment-4. “*Conclusion “that the artificially engineered cells can be developed as the novel living*

treatment materials for biomedical therapy” is too broad and too general and cannot be supported by existing data (which is related to resorbing calcified tissue in HO site).”

Response: Thanks for pointing out our inappropriate descriptions in the conclusion. In the new version of the manuscript, we have fixed the inappropriate descriptions that the referee reported. Now it reads (page 30, line 14), “Such an incorporation of functional materials into cells shows promise for cellular engineering by combining biology, chemistry and materials sciences to design artificially improved osteoclast as “living agents” for biomedical applications to HO reversible treatment.”

Comment-5. *“Methodology should be shortened.”*

Response: We have refined the Methodology section in the revised manuscript.

Comment-6. *“Thorough proof reading/editing is required, and English language usage should be improved.”*

Response: According to the comment, we polished the manuscript with a professional assistance in writing, conscientiously.

Comment-7. *“Errors in texts: Line 10 page 22: “First, OCs and immune cells are of bone marrow stromal cell origin”; Line 11 page 24: “Adherent cells were harvested as BMMs”.”*

Response: Thanks for the comment. We have corrected these errors and the improved sentences read

(page 29, line 4), “First, immune cells in the bone marrow interact with osteoclasts and osteoblasts through cell surface molecules, thereby regulating bone homeostasis.” and page 31, line 6, “BMMs were seeded onto 24-well plates (4×10^4 cells/well) or 96-well plates (1×10^4 cells/well) and cultured for 6 days in α -MEM supplemented with 10% (v/v) FBS, 25 ng/ml M-CSF and 50 ng/ml RANKL (R&D system) at 37 °C with 5% CO₂.”

Reviewers' Comments:

Reviewer #1:

Remarks to the Author:

This revised research report is much improved over the original submission. An important strength of this new report is the inclusion of multiple models for ectopic bone formation. However, there are three issues that need attention:

1. the repeated use of the phrase "gold standard" is not necessary.
2. This issue with the phrase "intelligent osteoclasts". Is not the word osteoclasts, it is the word intelligent. Osteoclasts do are not sentient beings. They do not have a brain.
3. In supplemental figures 13 and 14, no significance is indicated in the bar graphs. Is the reader to assume that none of the data is significant?

Reviewer #2:

Remarks to the Author:

The premise for the study is interesting and the authors have been responsive in trying to address reviewer comments and clarify their methods, data interpretation, and conclusions. However, the manuscript is marginally improved and remains difficult to understand sufficiently to agree with the authors' conclusions.

Reviewer #3:

Remarks to the Author:

The authors have done a good job in addressing my previous comments. I have no further comments/concerns.

Point-by-Point Response

Reviewer #1

Comment-1: *“The repeated use of the phrase "gold standard" is not necessary.”*

Response: Thanks for the comment! We have deleted two repeated descriptions of "gold standard" as the suggestion and now the two sentences read, *“To further prove the cells migrated to calcified tissue were mature osteoclasts, the calcified tissues were stained with TRAP, as the standard for osteoclast identification.”* (Page 7 line 4-6), *“The cytokine response of bone formation and resorption in the area of the cell implants was measured by ELISA. ALP, BMP-2, TRAP, CTX.”* (Page 11 line 20-21) and *“TRAP, type I collagen cross-linked C-telopeptides (CTX) were used as standard biomarkers for bone resorption.”* (Page 12 line 11-12).

Comment-2: *“This issue with the phrase "intelligent osteoclasts". Is not the word osteoclasts, it is the word intelligent. Osteoclasts do are not sentient beings. They do not have a brain.”*

Response: We accept this comment to remove the word of “intelligent”. Now, it reads, *“As natural bone resorbing cells, OCs can remove large amounts of calcium, phosphate and collagen fragments released by the dissolution of bone minerals and organic bone matrix during bone resorption, which would be toxic to cells.”* (Page 2, line 14-16)

Comment-3: *“In supplemental figures 13 and 14, no significance is indicated in the bar graphs. Is the reader to assume that none of the data is significant?”*

Response: Thanks for the advice! We are sorry for the negligence of unclear description the significance between different group in the previous version, which has been corrected in the present version. In Supplemental Figures 13 and 14, we provided such information in their legend as *“BMD after 2 weeks in Blank vs OCs: **p = 0.0014; Blank vs TC-OCs: ***p = 0.0003; OCs vs TC-OCs: p = 0.1656. BV/TV after 2 weeks in Blank vs. OCs: ns, p = 0.2081; blank vs. TC-OCs: **p =*

0.0071; OCs vs. TC-OCs: $*p = 0.0227$. ($*p < 0.05$, $**p < 0.01$, $***p < 0.001$)” and “BV/TV after 2 weeks in Blank vs. OCs: $*p = 0.0124$; blank vs. TC-OCs: $***p = 0.0006$; OCs vs. TC-OCs: $*p = 0.0324$ BS/TV after 2 weeks in blank vs. OCs: ns, $p=0.4982$; blank vs. TC-OCs: $**p=0.0095$; OCs vs. TC-OCs: $*p=0.0364$.” ($*p < 0.05$, $**p < 0.01$, $***p < 0.001$)”

Reviewer #2

Comment-1: “The premise for the study is interesting and the authors have been responsive in trying to address reviewer comments and clarify their methods, data interpretation, and conclusions. However, the manuscript is marginally improved and remains difficult to understand sufficiently to agree with the authors’ conclusions.”

Response: Thanks for the comment! The main conclusion of this study is that heterotopic ossification (HO) can be reversibly decalcified by using chemically modified osteoclasts (OCs). In order to enhance this new finding, we have performed the attempts on three representative types of heterotopic ossification (Achilles tendon transection, intramuscular and genetic models). The results can direct to that the chemically engineered osteoclasts can be extended their bone-resorption functions for the different heterotopic ossification treatments. The supports, especially the supports from the newly added experiments, are clearly and strong. Furthermore, this concept-of-proof study follows potential of artificially engineered cell in biomedical treatment, which is of great importance.

Reviewer #3

Comment-1: The authors have done a good job in addressing my previous comments. I have no further comments/concerns.

Response: We are happy to receive the approval and thanks for the great efforts on our manuscript improvement!